# Mouse papillomavirus type 1 (MmuPV1) DNA is frequently integrated in benign tumors by microhomology-mediated end-joining

**Lulu Yu**[1], **Vladimir Majerciak**[1], **Xiang-Yang Xue**[1], **Aayushi Uberoi**[2], **Alexei Lobanov**[3], **Xiongfong Chen**[3], **Maggie Cam**[3], **Stephen H. Hughes**[1], **Paul F. Lambert**[2], **Zhi-Ming Zheng**[1] *

**1** HIV Dynamics and Replication Program, Center for Cancer Research, National Cancer Institute, Frederick, Maryland, United States of America, **2** McArdle Laboratory for Cancer Research, University of Wisconsin School of Medicine and Public Health, Madison, Wisconsin, United States of America, **3** CCR Collaborative Bioinformatics Resource, National Cancer Institute, Bethesda, Maryland, United States of America

* zhengt@exchange.nih.gov

**Data Availability Statement:** NCBI GEO database Accession# GSE163537 for mouse tissue RNA-seq

## Abstract

MmuPV1 is a useful model for studying papillomavirus-induced tumorigenesis. We used RNA-seq to look for chimeric RNAs that map to both MmuPV1 and host genomes. In tumor tissues, a higher proportion of total viral reads were virus-host chimeric junction reads (CJRs) (1.9‰ - 7‰) than in tumor-free tissues (0.6‰ - 1.3‰): most CJRs mapped to the viral E2/E4 region. Although most of the MmuPV1 integration sites were mapped to inter-genic regions and introns throughout the mouse genome, integrations were seen more than once in several genes: Malat1, Krt1, Krt10, Fabp5, Pard3, and Grip1; these data were con-firmed by rapid amplification of cDNA ends (RACE)-Single Molecule Real-Time (SMRT)-seq or targeted DNA-seq. Microhomology sequences were frequently seen at host-virus DNA junctions. MmuPV1 infection and integration affected the expression of host genes. We found that factors for DNA double-stranded break repair and microhomology-mediated end-joining (MMEJ), such as H2ax, Fen1, DNA polymerase Polθ, Cdk1, and Plk1, exhibited a step-wise increase and Mdc1 a decrease in expression in MmuPV1-infected tissues and MmuPV1 tumors relative to normal tissues. Increased expression of mitotic kinases CDK1 and PLK1 appears to be correlated with CtIP phosphorylation in MmuPV1 tumors, suggest-ing a role for MMEJ-mediated DNA joining in the MmuPV1 integration events that are asso-ciated with MmuPV1-induced progression of tumors.

## Author summary

Persistent high-risk HPV infection leads viral DNA integration into the host genome and promotes viral carcinogenesis. We have been using the MmuPV1 mouse-infection model to study papillomavirus tumorigenesis and asked whether MmuPV1 DNA also integrates into the genomes of infected mouse cells. Strikingly, we found that MmuPV1 integration into the infected host genome, like high-risk HPV infections, is very common and the

and accession# GSE163681 for targeted mouse tissue DNA-seq.

**Funding:** ZMZ was supported by the Intramural Research Program of the National Institutes of Health, the National Cancer Institute, and the Center for Cancer Research (ZIASC010357). PL was supported by the National Institutes of Health grants (CA022443, CA210807, CA228543 and DE026787). The funders had no role in study design, data collection and analysis, decision to publish, or preparation of the manuscript.

**Competing interests:** The authors have declared that no competing interests exist.

mapped integration sites were distributed on all of the mouse chromosomes. Consistently, we identified microhomology sequences in the range of 2–10 nts always at the integration junction regions. We further verified the MMEJ-mediated viral DNA integration in tumor tissues during MmuPV1 infection and a step-wise increase in the expression of the DNA repair MMEJ host factors from normal tissues, to tumor-free MmuPV1 infected tissues, and then to MmuPV1 tumors. Our observations provide the first evidence of MmuPV1 integration in virus-infected cells and a conceptual advance of how papillomavirus DNA integration contributes to the development of papillomavirus-associated pre-cancers to cancers.

## Introduction

Papillomavirus genomes replicate as extrachromosomal DNA circles in the nucleus; integration of the viral DNA into the host genome is a dead end for the virus. This is in contrast to retroviruses, like HIV, for which integration of a DNA copy of the viral genome into the host genome is a critical step in the viral replication cycle that has important implications for the persistence of the HIV during successful anti-viral therapy [1, 2]. Human papillomaviruses (HPV) is the main cause of cervical cancer [3] and numerous studies have shown that integrated HPV DNA is present in many HPV-related cancer tissues, including cervical cancer [4, 5], head and neck cancer [6, 7] and oropharyngeal cancer [8]. Although papillomaviruses do not deliberately integrate their genomes, integrated HPV DNA was shown to be present in HPV18 positive cervical cancer cell lines in 1985 [9]. The number of integrated HPV copies correlates with the severity of cervical lesions [10, 11] and is a biomarker for disease progression. In the integrated copies of HPV DNA in tumor samples, the break sites in the viral genome are commonly found in the E1 and E2 genes [12, 13]. Loss of E2 expression from the integrated DNA leads to increased expression of the viral oncoproteins E6 and E7 that are essential for the proliferation and survival of HPV-related cancer cells [14–16]. Thus, the fact that HPV DNAs found in tumors are often integrated in E1 and E2 represents a selection for forms of integrated HPV DNA that can help transform the host cell, and is not caused by a propensity for HPV DNA to break and/or integrate in specific regions. Integrated HPV DNA can be found in many places in the host genome. HPV integration can affect the expression of the MYC oncogene [17, 18] or form a viral-cellular super-enhancer that promotes viral oncogene expression [19], although the latter event is uncommon. The mechanisms that are involved in the integration of HPV DNA into the host genome remains unclear. However, high-risk HPV infection induces a DNA damage response that is important for viral DNA replication, but causes host genomic instability [20–24]. Microhomology-mediated end-joining (MMEJ) is a double-stand break (DSB) repair mechanism that uses short regions of homologous DNA sequences ($> = 1$ bp) for error-prone end-joining [25–27]. MMEJ has been found in the integration sites of several pathologic viruses, including adeno-associated virus, Merkel cell polyomavirus (MCV) and papillomavirus [7, 13, 28, 29].

Papillomaviruses are species-specific, which makes it difficult to study HPV integration because there is no good animal model. Mice can be infected with mouse papillomavirus type 1 or Mus musculus papillomavirus 1 (MmuPV1) in both mucosal and cutaneous sites [30–32]. MmuPV1 is transmissible through blood [33] and sexual contact [34] and induces warts that are similar to pre-cancerous stages caused by HPV infections in humans. Like HPV infections, MmuPV1 infections can cause cancer [32, 35], making it a good model for human cancer studies [36, 37]. However, there are no published reports on MmuPV1 integration. We asked if

MmuPV1 DNA integrates into the genomes of infected mouse cells and if MmuPV1 integration is similar to the HPV DNA integration.

Both DNA-based and RNA-based assays have their advantages and limitations in studying HPV and MmuPV1 DNA integration events. Theoretically, DNA-based assays are capable of detecting all of the viral-host junctions, but the throughput and the sensitivity of the assays are limited [38]. RNA-based assays are more sensitive than the DNA-based assays but only detect integrated viral DNA if the region that contains the viral DNA is transcribed; integration sites that are transcriptionally silent are missed. If the host-virus RNA chimera is spliced, the exact junction site can be lost. Integrated viral DNA was first analyzed by PCR-based methods, such as Detection of Integrated Papillomavirus Sequences by ligation-mediated PCR (DIPS-PCR) and Amplification of Papillomavirus Oncogene and Transcripts (APOT) [39, 40]; however, whole genome DNA sequencing (WGS) and RNA sequencing (RNA-seq) using the Illumina platform are now more widely used. RNA-seq is more sensitive than WGS because active genes express multiple copies of their RNA transcripts. The Illumina platform produces millions of short reads (75 bp-150 bp). In contrast, single-molecule sequencing technology (SMRT-seq) [41] (Pacific Biosciences (PacBio)) produces longer reads (up to >50 kb) that allow better mapping, although the throughput is lower and the number of reads is smaller. In the current study, we combined RNA-seq, SMRT-seq and targeted DNA-seq to detect MmuPV1 integration sites in MmuPV1-infected tissues and tumors. We provide direct evidence that the viral DNA is integrated in cells infected by MmuPV1.

## Results

### RNA-seq analysis of integrated MmuPV1 DNA in MmuPV1-induced tumor tissues

Persistent MmuPV1 infection induces papillomas (warts), which are benign tumors. **Fig 1A** shows a benign tumor with an inner stromal layer, an intermediate layer of hyperplastic and koilocytotic keratinocytes, and an outer layer of terminally differentiated and cornified cells. The life cycle of MmuPV1 is similar to the life cycle of all other papillomaviruses and is tightly linked to keratinocyte differentiation. The viral early genes (E2, E6 and E7) are expressed in the undifferentiated cells and the viral late genes (L1 and L2) are expressed only in highly differentiated cells in the outer layer (**Fig 1B**) of the tumor with atypical epidermal hyperplasia, koilocytes and hyperkeratosis (**Fig 1C**).

We used RNA-seq analysis to search for MmuPV1 integration sites, in both the viral and mouse genomes, in nine MmuPV1-infection-induced tumors (warts) and three tumor-free MmuPV-1-infected tissues [42]. We defined two types of paired-end reads as virus-host chimeric reads: chimeric paired reads and chimeric junction reads (CJRs) (**Fig 2A**). The proportion of virus-host chimeric reads is much higher in tumor tissues (1.9‰ - 7‰ of total viral reads) than in infected, tumor-free tissues (0.6‰ - 1.3‰ of total viral reads) (**Fig 2B**). For example, although tumor tissue sample #5 had a slightly lower number of viral reads (949,720 reads) than non-tumor tissue sample #10 (971,642 reads), the proportion of its virus-host chimeric reads was 10 times higher in the tumor tissue (6.4‰) than in the tumor-free tissue (0.6‰). This result is consistent with reports that, for high-risk HPV, high levels of integrated DNA correlates with the grade of cervical lesions [10, 11].

If the E2 gene is disrupted by HPV integration, the expression of the viral oncogenes E6 and E7 is increased because transcriptional suppression by E2 is lost [43]. When RNA-seq was used to map the virus-host CJRs to MmuPV1 genome, the majority of the CJRs were within the E2 gene and the overlapping E4 gene (**Fig 2C**). We previously showed that when MmuPV1-specific RNAs were mapped on the viral genome, there was a marked reduction in

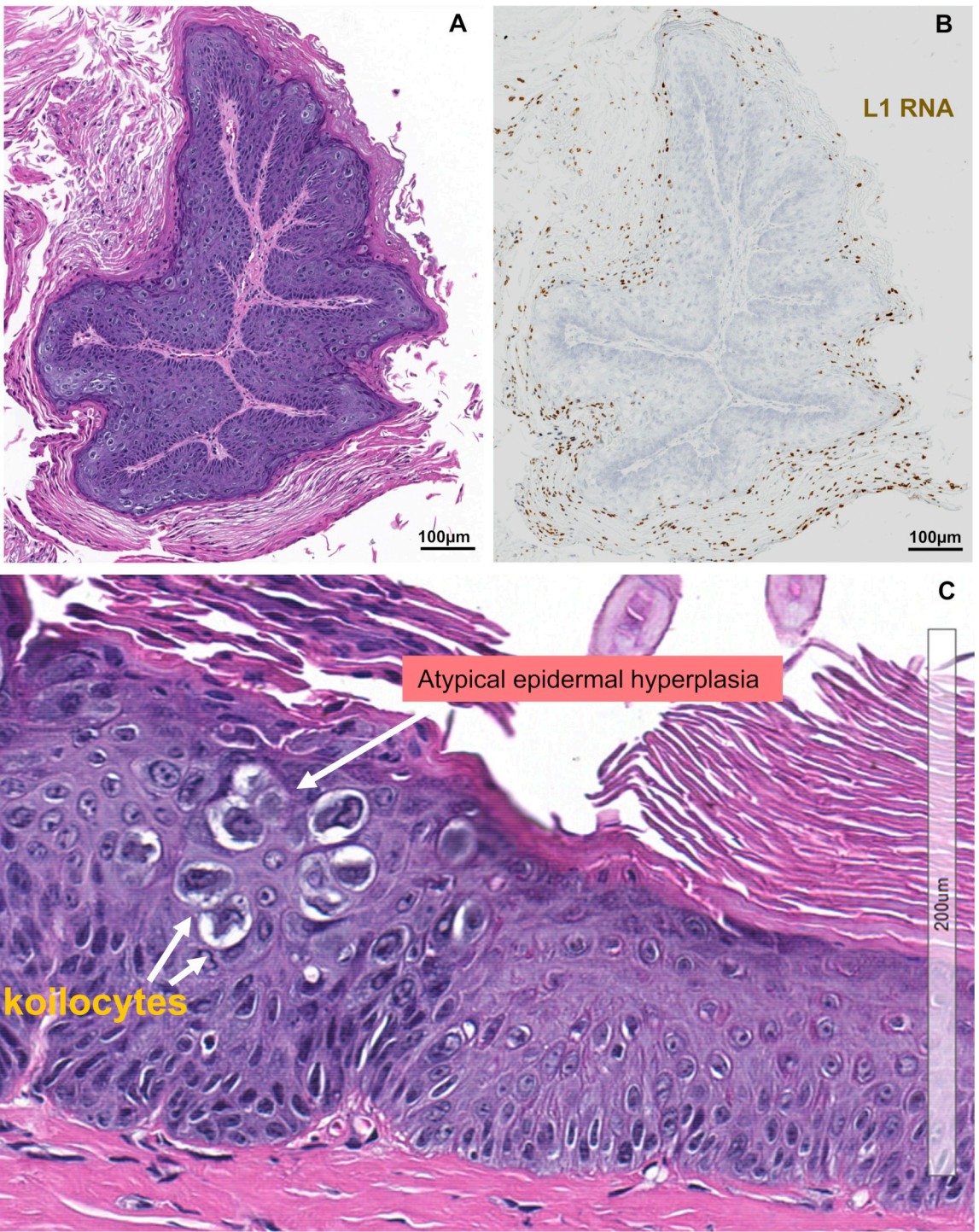

**Fig 1. MmuPV1 induced papillomas express viral L1 RNA. A,** H&E staining of a thin section of a classic papilloma showing the inner layer stroma, intermediate layer of hyperplastic and koilocytotic epithelial cells, and outer cornified, keratinized layer. This tissue was collected from a female mouse muzzle tumor after six months of MmuPV1 inoculation. **B,** RNA *in situ* hybridization (RNA-ISH) of MmuPV1 L1 from a serial section of the same MmuPV1-infected papilloma which was detected by an RNAscope assay with a MmuPV1-specific L1 probe. The tissue was DNase-treated to remove all viral DNA before hybridization. This image shows that MmuPV1 L1 RNA (yellow brown color) is preferentially expressed in the terminally differentiated keratinocytes and the keratinized cells in the outer layer. **C,** An enlarged area of the section in panel A showing atypical epidermal hyperplasia, koilocytes and hyperkeratosis.

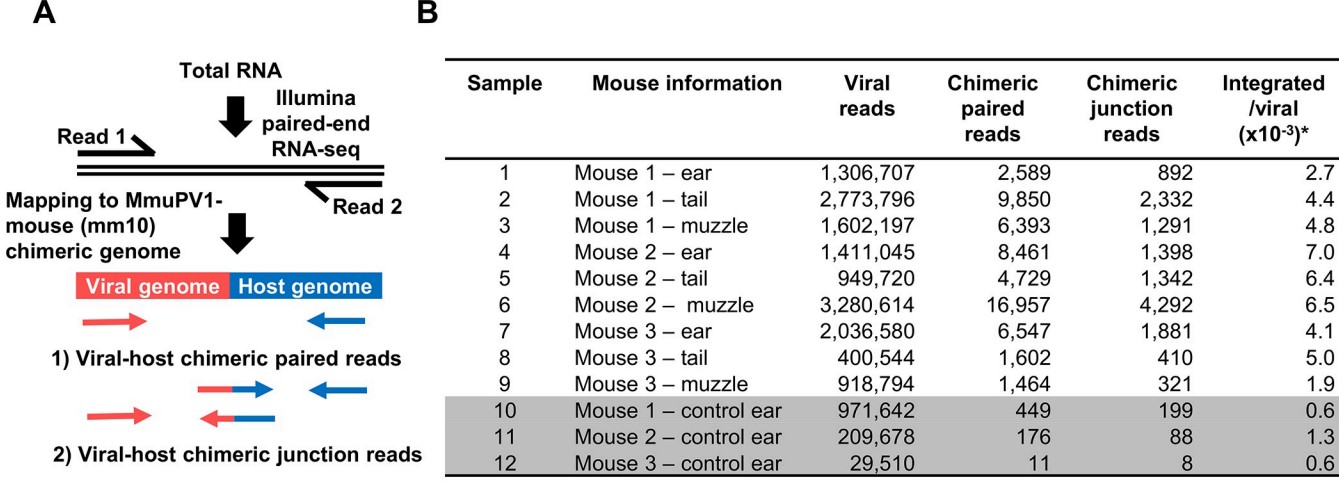

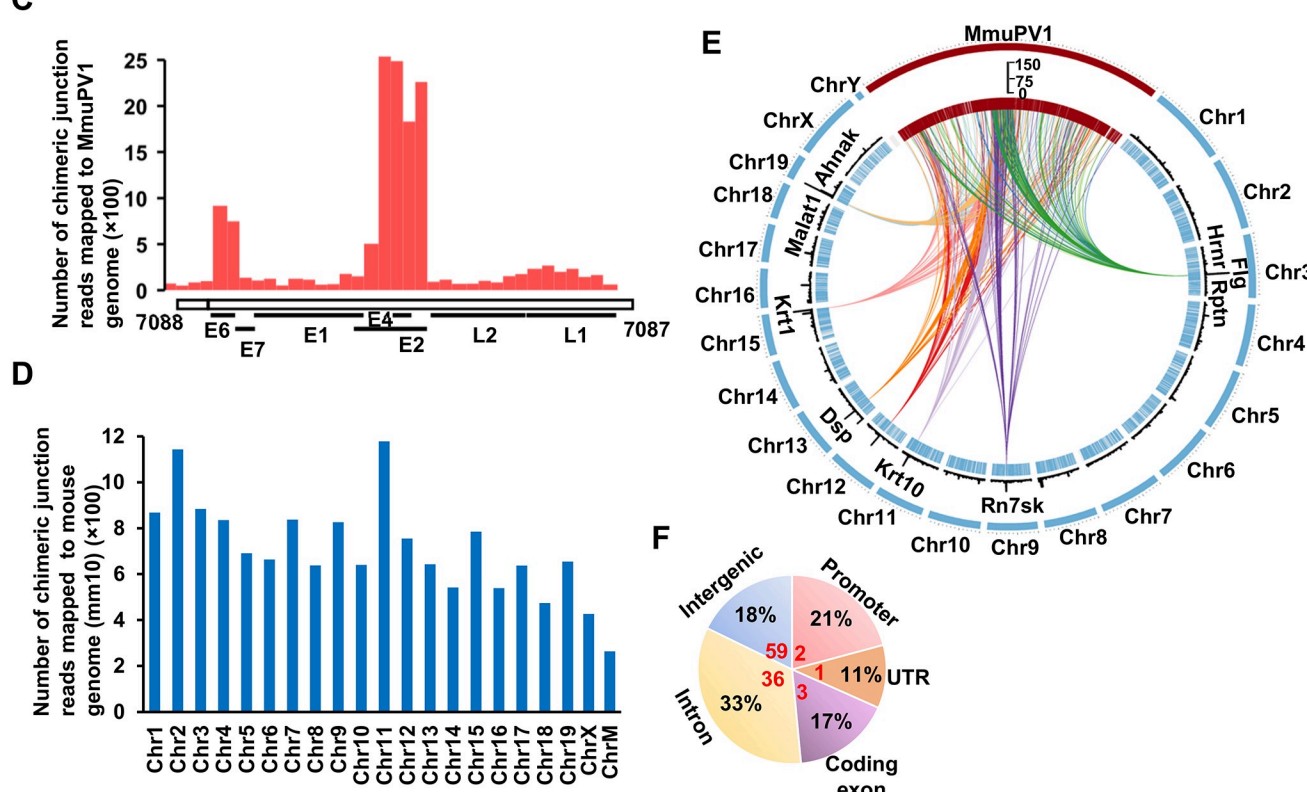

| Sample | Mouse information | Viral reads | Chimeric paired reads | Chimeric junction reads | Integrated /viral (x10⁻³)* |
|--------|-------------------|-------------|----------------------|------------------------|---------------------------|
| 1 | Mouse 1 – ear | 1,306,707 | 2,589 | 892 | 2.7 |
| 2 | Mouse 1 – tail | 2,773,796 | 9,850 | 2,332 | 4.4 |
| 3 | Mouse 1 – muzzle | 1,602,197 | 6,393 | 1,291 | 4.8 |
| 4 | Mouse 2 – ear | 1,411,045 | 8,461 | 1,398 | 7.0 |
| 5 | Mouse 2 – tail | 949,720 | 4,729 | 1,342 | 6.4 |
| 6 | Mouse 2 – muzzle | 3,280,614 | 16,957 | 4,292 | 6.5 |
| 7 | Mouse 3 – ear | 2,036,580 | 6,547 | 1,881 | 4.1 |
| 8 | Mouse 3 – tail | 400,544 | 1,602 | 410 | 5.0 |
| 9 | Mouse 3 – muzzle | 918,794 | 1,464 | 321 | 1.9 |
| 10 | Mouse 1 – control ear | 971,642 | 449 | 199 | 0.6 |
| 11 | Mouse 2 – control ear | 209,678 | 176 | 88 | 1.3 |
| 12 | Mouse 3 – control ear | 29,510 | 11 | 8 | 0.6 |

* Student's *t*-test P < 0.01 between tumor and tumor-free control groups.

**Fig 2. RNA-seq analyses of MmuPV1 integrations in mouse tumor and tumor-free tissues. A,** Diagrams of the two principal types of chimeric RNA reads that derive from a viral integration site. Chimeric paired-end reads obtained in RNA-seq analysis were mapped to the MmuPV1 reference genome (NC_014326) and the mouse reference genome (mm10). The virus-host chimeric paired reads are defined by one read that fully maps to the virus genome and the other read that fully maps to the host genome. A virus-host chimeric junction read (CJR) is a read that contains the virus-host junction that links sequences in the virus genome and the host genome independent from RNA splicing. **B,** The number of chimeric RNA reads from tumor tissues from three distinct MmuPV1-inoculated anatomical sites (ear, tail and muzzle) of three separate female mice. The left ear (without MmuPV1 inoculation) was from the same animal, which had a normal appearance, and served as a normal control for total RNA extraction and RNA-seq analysis. **C,** Distribution of mapped integration breakpoints (CJRs) across the MmuPV1 genome in nine tumor samples. **D,** Chromosomal distribution of the mapped integration sites in the mouse genome based on the CJRs ChrM, mitochondrial chromosome. **E,** Distribution of the MmuPV1 integration events across the mouse genome. The outer circle represents the locations of the viral genome (MmuPV1, red) and the 20 mouse chromosomes (chr, blue). The sizes of mouse chromosomes are proportional to their lengths, and the viral genome is scaled up 100,000x. Each "tick" in internal circle indicates a detected integration event. The black bars in the middle depict the frequency of the MmuPV1 integration events in each region and the 9 genes with the highest number of integration sites are highlighted. The circular diagram was created using Circos package (doi:10.1101/gr.092759.109). **F,** Pie chart showing percent distribution (% in black) of the mapped RNA CJRs in the different genomic regions. Numbers in red are the normal fraction (%) of each region in the mouse genome. The CJRs (integration sites) were annotated using the annotate Peak command of ChIPSeeker (doi:10.1093/bioinformatics/btv145) package.

the RNA reads that mapped to the E2/E4 region of the viral genome in warts [42]. We speculated that there were fewer RNAs that map to this region because this region is frequently interrupted in integrated copies of the viral genome. The E2/E4 region is part of the 5' non-coding region for L1 mRNA [44]. Thus, if integration of the viral genome disrupts this region, the expression of L1 RNA and L1 protein would be blocked.

We mapped all virus-host CJRs identified by RNA-seq to the mouse genome. After excluding the 86 CJRs that were derived from RNA splicing, we identified 14,159 CJRs from nine tumors and 295 CJRs from tumor-free tissues. The mapped integration sites were distributed on all of the mouse chromosomes (**Fig 2D and 2E**), with the RNA CJRs mapping more frequently to the promoter, UTR and exon regions and less frequently to the intergenic regions, relative to the fractions these regions represent in the mouse genome (**Fig 2F**). There were multiple integrations in Malat1, Flg, Krt1, Dsp, Krt10, Hrnr, and Rn7sk (**Fig 2E and S1 Table**).

## Verification of RNA-seq integration site data by RACE-SMRT-seq

To confirm the presence of integration sites in the host genes with multiple MmuPV1 integration sites identified by RNA-seq, we performed 5' rapid amplification of cDNA ends (5' RACE) using total RNA (1 μg) extracted from the tumor sample #6 and 3' RACE using total RNA (1 μg) extracted from the pooled tumor samples #2, #4 and #6, followed by Single Molecule Real-Time sequencing (SMRT-seq) (**Figs 3A and S1A and S1B**). SMRT-seq of the MmuPV1-specific primer-derived 5' and 3' RACE products yielded 280 and 694 clustered chimeric long reads, respectively. We mapped these clustered virus-host chimeric long reads, respectively, to 97 and 529 annotated mouse genes (**S2 and S3 Tables**). Fewer long reads were recovered because SMRT-seq produced fewer reads (~40,000) than RNA-seq (~100 million). As expected, most of the chimeric junctions (breakpoints) in the long reads mapped to the E2/E4 region of the MmuPV1 genome (**S1C and S1D Fig**). Genes that had multiple integration sites in the SMRT-seq analysis were Krt10, Krt6, Malat1 and Dsp (**S1E and S1F Fig**). Using RACE-SMRT-seq, we were able to verify MmuPV1 integrations in more than 73% of the top 15 genes (11/15) that were identified by RNA-seq (**Fig 3B**). The lengths of these genes ranged from 0.3 kb—87.6 kb.

The Krt10 gene was one of the genes in which multiple integrations were identified in MmuPV1-induced tumor tissues by both RNA-seq and RACE-SMRT-seq. As shown in **Fig 3C**, in the viral DNAs that were integrated into exon 1 of Krt10, the viral genome was linearized within E2 region (also see **Figs 2C and S1C and S1D**). The chimeric full-length viral-Krt10 transcripts detected by a specific viral primer are represented in **Fig 3C**, showing the three chimeric RNAs transcribed from the Krt10 promoter that were detected by 5' RACE-SMRT-seq and the three chimeric RNAs that were transcribed from the viral promoters that were detected by 3' RACE-SMRT-seq (**S2 and S3 Tables**). Using the MmuPV1-Krt10 full-length transcript C8112 as a template, we were able to map the RNA-seq generated short reads, including the short CJRs, to this template (**Fig 3D**). The data clearly show that MmuPV1 integrations in the Krt10 exon 1 region can be mapped using high-throughput technologies.

## Targeted DNA-seq of MmuPV1 DNA integration sites

We performed targeted DNA-seq to identify additional MmuPV1 integration sites. DNA was prepared from MmuPV1-induced tumor (ear and muzzle) and MmuPV1-infected, tumor-free (ear) tissues from the same mouse #2 (**Fig 2B**) to avoid animal to animal variation. We showed, using ddPCR, that there were 6,630 and 34,853 copies of the viral genome per cell in ear tumor and muzzle tumor, respectively, and 2,547 viral genome copies per cell in a tumor-free ear tissue (**S2A Fig**). We used gel electrophoresis to separate the extrachromosomal viral DNA from the chromosomal DNA (**Fig 4A, gel panel**). The chromosomal DNA that was

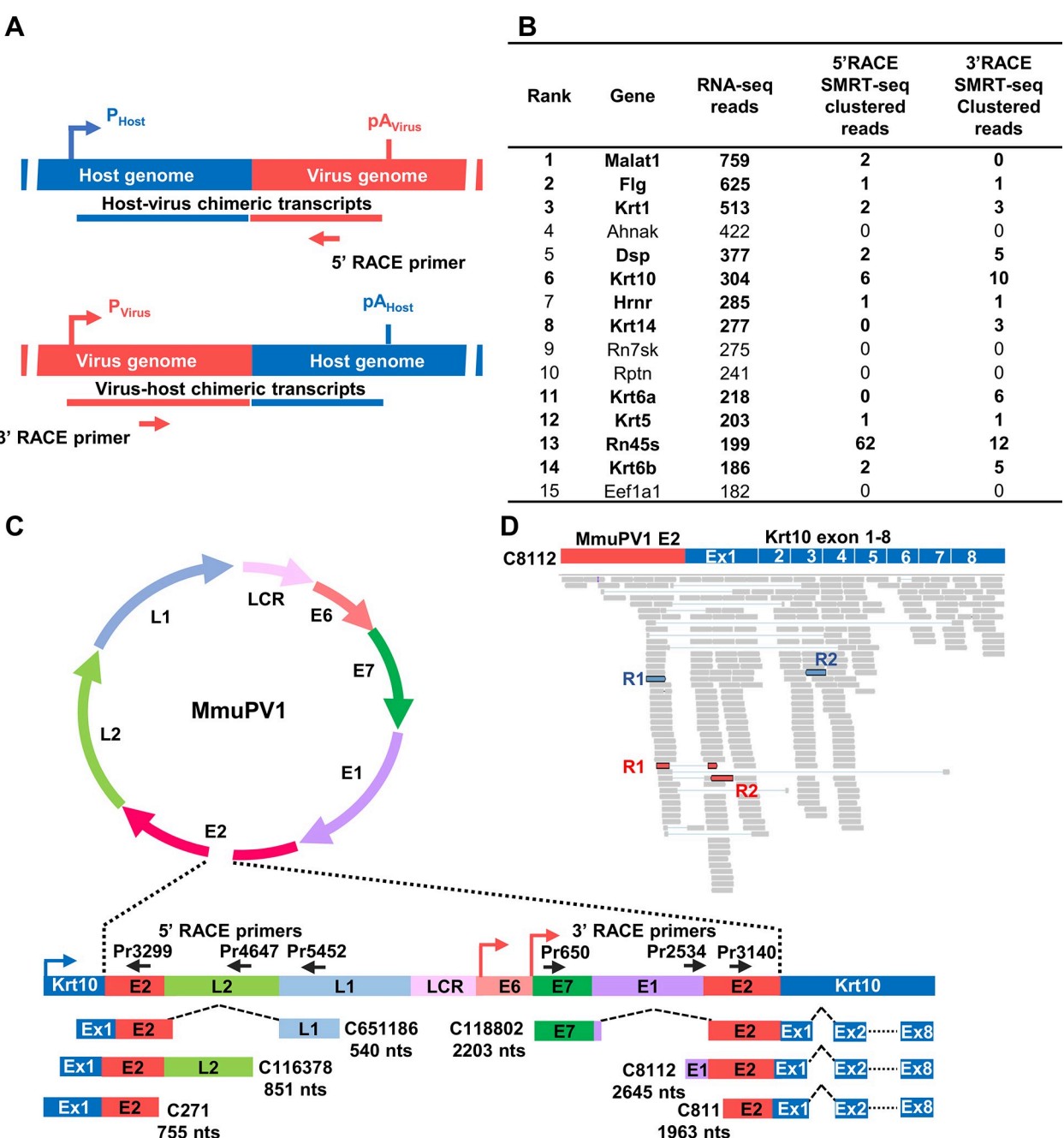

**Fig 3. Validation of MmuPV1 integration sites by 5' and 3' RACE in combination with SMRT-seq (single molecule real-time sequencing). A,** Strategies used to validate MmuPV1 integration sites by 5' RACE and 3' RACE with virus-specific primers. The chimeric virus-host cDNA amplicons derived from both 5' and 3' RACE were sequenced by SMRT-seq. $P_{virus}$ or $P_{host}$ indicates the use of a viral or host promoter and $pA_{host}$ or $pA_{virus}$ indicates the use of a host or viral polyadenylation signal. **B,** Verification by RACE-SMRT-seq of the 15 most favored host genes with MmuPV1 integration (S1 Table) detected by RNA-seq. Genes detected by two methods are bolded. **C,** Examples of virus-host (Krt10) full-length chimeric transcripts detected by 5' RACE-SMRT-seq and 3' RACE-SMRT-seq. Virus primers (black arrows) used in the RACE reactions are shown above the linearized and integrated viral genome, with colored arrows for host (blue) and viral (pink) promoters. Below are the detected chimeric, full-length virus-host RNA transcripts, with the name and estimated size in nts (after RNA splicing) given on the right or left side. **D,** RNA-seq reads-coverage for the virus-Krt10 chimeric transcript C8112 identified by 3' RACE-SMRT-seq and visualized by IGV software. The representative chimeric junction reads (red) and chimeric paired reads (blue) from the RNA-seq data are highlighted.

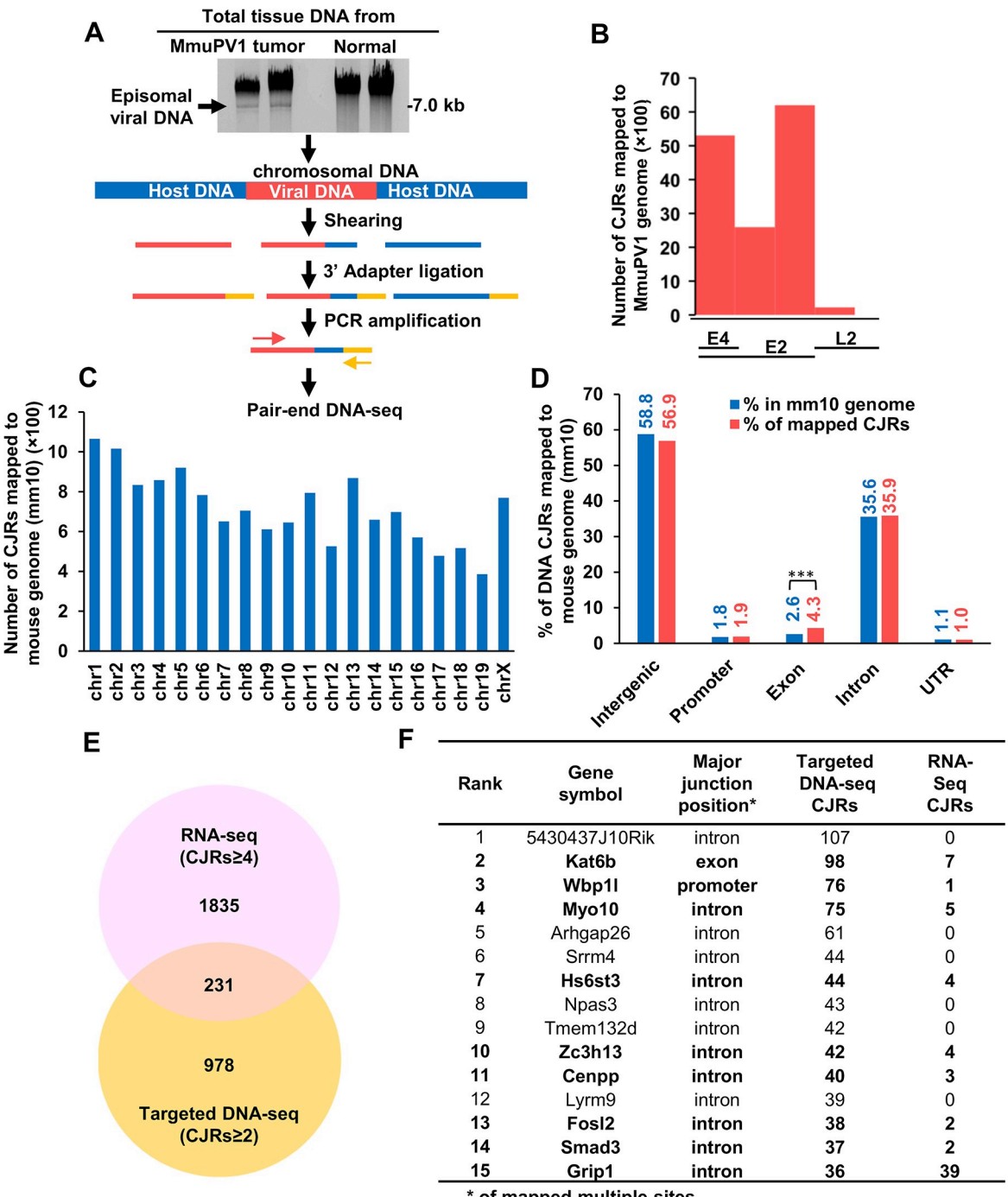

**Fig 4. Detection of MmuPV1 integration sites in MmuPV1 tumor tissues by targeted DNA-seq. A,** Schematic presentation of the strategies for detecting the integration sites for MmuPV1 DNA in chromosomal DNA by targeted DNA-seq. Tumor or normal ear tissues of animal #2 (Fig 2B) were dissected and used for genomic DNA extraction. The episomal viral genomes were separated from the chromosomal DNA by agarose gel electrophoresis and the genomic DNA was isolated from the gel and sheared by sonication. The sheared DNA fragments were end-repaired and their 3' ends were ligated to an adaptor. Three specific viral primers (Pr2976, Pr3277 and Pr3742) at MmuPV1 E2 region (S10 Table) were used separately, in combination with an adapter primer, to selectively PCR amplify chimeric virus-host DNA fragments, and the amplicons were then sequenced. **B,** Distribution of integration breakpoints in the MmuPV1 genome by mapped CJRs from targeted DNA-seq. **C,** Chromosomal distribution of the MmuPV1 integration sites in the mouse genome by mapped CJRs. **D,** The proportion of different host genomic regions with MmuPV1 integration sites. ***, P<0.001 in Chi-squared test. **E,** Genes with MmuPV1 integration sites that were identified by RNA-seq (CJRs ≥ 4) and verified by targeted DNA-seq (CJRs≥ 2) are shown in a Venn diagram. **F,** The top 15 genes with the most MmuPV1 integration sites identified by targeted DNA-seq and RNA CJRs detected by RNA-seq.

recovered from the gel was fragmented by shearing, end-repaired, and dA-tailed before being ligated to a double-stranded 3' oligo adaptor containing a 3' T overhang. The viral-host chimeric fragments were selectively amplified using an adaptor-specific primer in combination with one of three forward, MmuPV1-specific, E2 primers designed based on our RNA-seq results (see **Fig 4A flowchart**). This approach allowed us to selectively detect one of the two junctions for each of the integrated MmuPV1 DNAs that were broken in E2. As expected from viral genome copy number results (**S2A Fig**), DNA-seq using the same amount of DNA for each sample showed that there were two times more virus-host DNA CJRs (7,007 CJRs) in the MmuPV1 ear tumor than that (2,753 CJRs) in MmuPV1-infected, tumor-free ear (**S2A Fig**). However, we only detected 4,593 DNA CJRs in the muzzle tumor despite the fact that this tumor tissue had 34,853 copies of the viral genome per cell, suggesting there were many more extrachromosomal viral DNA copies in the muzzle tumor than in the ear tumor tissue. As expected from the RNA-seq analysis data and our targeted DNA-seq strategy, most of the DNA CJRs amplified were mapped to the viral E2/E4 region, only a small number of CJRs were from the L2 region (**Fig 4B**) downstream of E2. The targeted DNA-seq amplification was able to pick up integration sites in L2 because L2 is near E2.

Mapping the CJRs from all three samples to the mouse genome showed that integrated MmuPV1 DNA was widely distributed throughout the genome (**Fig 4C**). We found ~93% of the MmuPV1 integration sites were present in intergenic and intronic regions of the mouse genome. Although only 4% of the integration sites were in the coding exons, this is more than was expected (P<0.001) (**Fig 4D**) because the coding regions make up only ~2.6% of the genome (www.ncbi.nlm.nih.gov/assembly/GCF_000001635.20/). In the intergenic regions, there appear to be more than the expected number of viral integration sites in the long interspersed nuclear elements (LINEs) (**S2B Fig**) (P<0.01), which is similar to what has been reported for HPV integration [13]. Among 1,209 host genes with CJRs identified by targeted DNA-seq, we found virus-host RNA CJRs in 231 (19.1%) by RNA-seq (**Fig 4E and S4 Table**). Among the top 15 of 100 genes (**S5 Table**) with multiple MmuPV1 integration sites identified by targeted DNA-seq, we identified, by RNA-seq, expressed chimeric RNA transcripts in nine (60%) (**Fig 4F**).

Among the 1,048 host genes with ≥ 2 DNA CJRs, 569 were identified from ear tumors, 282 from muzzle tumors, and 197 from MmuPV1-infected, tumor-free ears (**Fig 5 and S6 Table**). We obtained, from an MmuPV1 induced ear tumor, 487 more genes with MmuPV1 integrations than were obtained from the MmuPV1-infected, tumor-free ear. There were 40 genes in which we detected MmuPV1 integrations in both ear and muzzle tumors but not in the tumor-free tissue (**Fig 5**), which suggests that there might have been post-integration selection for the cells with MmuPV1 integration in these genes.

## MmuPV1 integration can affect the expression of the host genes

Although RNA-seq has been used to profile the differential expression of host genes in MmuPV1-infected cells, it was difficult to directly determine the effects of MmuPV1 integration on host gene expression because the populations of infected cells with MmuPV1 DNA integrations are very heterogeneous and some fraction of the MmuPV1-infected cells in the papilloma tissues may not have any integrated MmuPV1 DNA. On the other hand, majority of the integrated MmuPV1 DNA that was integrated in the intergenic regions (**Fig 4D**) were not transcribed (**Fig 2F**). This is presumably due to the lack of a nearby host promoter in the intergenic regions. The MmuPV1 DNAs that are integrated in genes are much more likely to be detected in the chimeric RNA analyses. We analyzed the relative expression of the 40 genes with MmuPV1 integrations identified by targeted DNA-seq from two different tumor sites

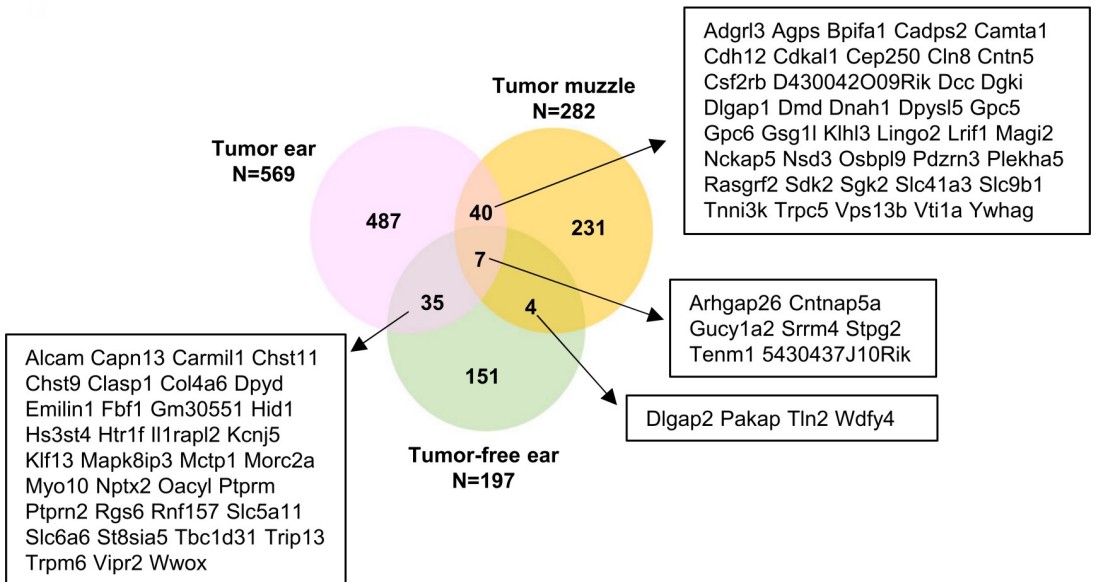

**Fig 5. Identification of the common genes with MmuPV1 DNA integration in two different tumor sites from the same animal.** A Venn diagram showing the number of genes with multiple MmuPV1 integration sites (CJR≥ 2) in three different tissues from animal #2 (**Fig 2B**) identified by targeted DNA-seq.

from the same animal (**Fig 5**). We first compared the RNA-seq data from three ear tumors (**Fig 2B**, **samples #1, #4 and #7**) and three MmuPV1-infected, tumor-free ears (samples #10, #11 and #12) from the same 3 animals (**S7 Table**), and then tumor ear (sample #4) vs tumor-free ear (sample #11) from the mouse #2 (**S8 Table**). We found that there was a significant (P<0.05) differential expression (≥ 1.4 or ≤ -1.4-fold, RPKM ≥ 0.5) of 8 genes in the ear tumors, of which one (Cln8) had a viral DNA integrated in a 6-kb terminal exon and was upregulated. Seven genes (Pdzrn3, Csf2rb, Dmd, Plekha5, Camta1, Gpc6 and Rasgrf2) which had viral DNA integrated in their introns were downregulated (**S8 Table**). We also found RNA CJRs that involved the Cln8, Pdzrn3, Camta1 and Plekha5 genes (**S4 Table**), but did not find CJRs for Csf2rb, Dmd, Gpc6 and Rasgrf2.

We found multiple MmuPV1 integration sites in genes that were either highly expressed or weakly expressed (**Fig 6A and S7 Table**). For example, the Malat1 and Krt10 genes had hundreds to thousands of detectable RNA reads and the Par-3 family cell polarity regulator or partitioning defective 3 homolog (Pard3) and glutamate receptor interacting protein 1 (Grip1) had only a few (<20) detectable RNA reads. We were unable to detect a change in the expression of these host genes when the tumor samples were compared to tumor-free, MmuPV1-infected tissues.

We also analyzed the expression of Krt10 and Fabp5, two highly expressed genes that had multiple MmuPV1 integration sites in tumor tissues and performed RNA ISH using RNA-scope technology. We found that there was a reduction of both Krt10 and Fabp5 RNA levels in the MmuPV1-induced tail tumors 21 days after viral infection. This reduction was associated with a high level of viral E6/E7 RNA compared with the tumor-free areas that had little or no viral E6/E7 RNA (**Fig 6B and 6C**). Quantitative analysis done by hybridization to the Krt10 and Fabp5 RNAs showed that there was a 41.4% reduction of Krt10 and 38% reduction of Fabp5 in the regions of the tumors that expressed E6/E7 compared to the adjacent normal tissues that did not express E6/E7 (**Fig 6B and 6C, bar graphs**). These data suggest that the

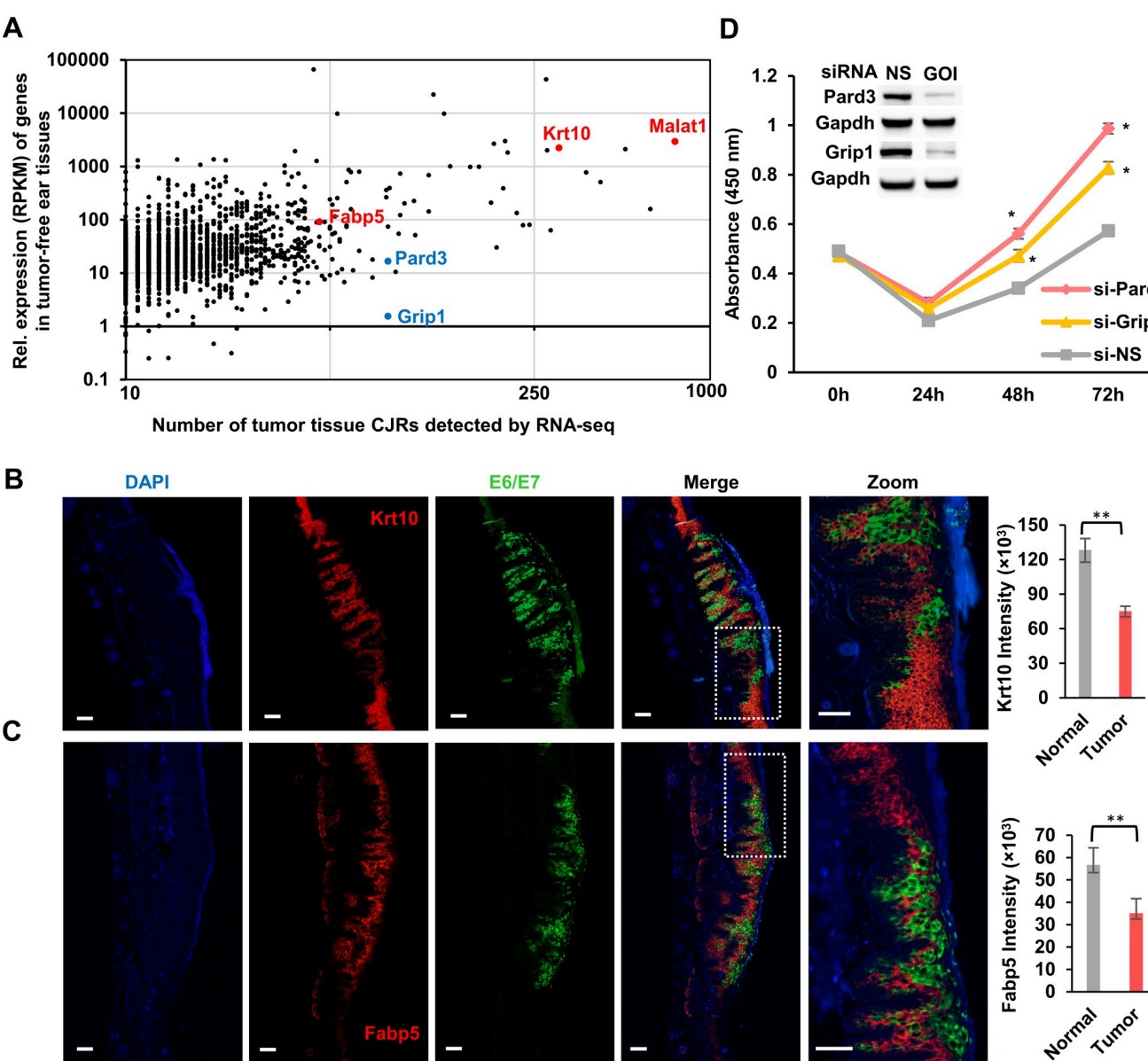

**Fig 6. MmuPV1 integration, host gene expression, and cell growth. A,** MmuPV1 integration sites were found in genes that are expressed at both high and low levels. Each dot indicates a gene with the indicated level of expression (RPKM) in tumor-free ear tissues and the number of virus-host CJRs obtained from all tumor tissues by RNA-seq. RPKM is the Reads Per Kilobase of transcript per Million mapped reads in RNA-seq. **B and C,** Expression of MmuPV1 E6E7 (green) correlates with a decrease in the expression of mouse Krt10 and Fabp5 (red) in the tail tumor tissues of FoxN1[nu/nu] at day 21 as measured by RNA ISH using dual immunofluorescence RNAscope analysis. Nuclei were stained with DAPI (blue). Scale bar, 100 μm. Krt10 and Fabp5 levels in four randomly selected tumor tissues (that express E6E7) and four tumor-surrounding normal tissues (that do not express E6E7) were quantified by Image J, averaged, and shown as a bar graph on the right. Statistical comparisons were computed by unpaired, two-tailed Student's $t$ test. ** $p < 0.01$. **D,** Knockdown of Pard3 or Grip1 expression by gene-specific siRNA promotes cell proliferation. The siRNA knockdown of Pard3 and Grip1 expression in mouse primary keratinocytes was confirmed by Western blot, with Gapdh serving as a sample loading control. Primary mouse keratinocytes were transfected with 40 nM of siRNA twice at a 24 hr-interval and analyzed by a WST-8 cell proliferation assay. NS, non-specific siRNA; GOI, gene of interest. Data are the mean ± SE of three repeats. Statistical comparisons were computed by unpaired, two-tailed Student's $t$ test. * $p < 0.05$.

reduction in the levels of Krt10 and Fabp5 RNAs is a consequence of the loss of cellular differentiation and is not direct consequence of viral DNA integration.

The Pard3 and Grip1 genes were expressed at low levels; however, we were able to recover MmuPV1 integration sites in these genes by both RNA-seq and targeted DNA-seq (**Fig 6A**

and **S4** and **S7** Tables). Pard3 controls cell polarity and plays an important role in epithelial cell tumorigenesis [45]. Grip1 is an adaptor protein composed of seven PDZ domains that participates diverse protein-protein interactions [46]. Using siRNAs to knock down the expression of Pard3 and Grip1 in mouse primary keratinocytes significantly promoted cell proliferation (**Fig 6D**). Knocking down Pard3 had a greater effect than knocking down Grip1. Flow cytometry shows that knockdown of Pard3 and Grip1 was associated with an increase in cell cycle transition (**S3 Fig**). We found that knockdown of Pard3 reduced the number of cells in G0/G1 and increased cell number of entering G2/M, whereas knockdown of Grip1 accelerated the transition from G0/G1 to S (**S3 Fig**).

## Identification of microhomology sequences at MmuPV1 integration sites

Microhomology sequences (MHS) are present at aberrant HIV-1 integration sites that were not mediated by integrase (IN) [47–50] and at the integration sites of adeno-associated virus, papillomavirus and MCV [13, 28, 29]. We analyzed the virus-host junction sequences by both RNA-seq and targeted DNA-seq by comparing the sequences of the MmuPV1 genome and the mouse genome. As discussed earlier, the CJRs identified by RNA-seq in which the junctions were mediated by virus-host RNA splicing were excluded from the analysis. We found MHS in the range of 2–10 nts at the junction region of most of the CJRs. MHS, the defining hallmark of microhomology-mediated end-jointing (MMEJ) [25, 26], were significantly enriched in both data sets (**Fig 7A and 7B**). As shown in **Fig 7C,** MmuPV1 MHS of 4 nts were found for Krt10, 5 nts for Malat1 and Grip1 in the virus-host junctions identified by RNA-seq.

We used nested RT-PCR to confirm the MHS in the chimeric junctions of MmuPV1-Krt10 transcripts in tumor tissues based on the position of CJRs identified by RNA-seq. As shown in **Fig 7D**, RT-PCR, using primer pairs for Krt10 in combination with MmuPV1, we detected a chimeric MmuPV1-Krt10 RNA of the expected size from total RNA extracted from a MmuPV1 tumor tissue (**Fig 7D gel panel, lane 3**), but not a negative control Gapdh RNA (**Fig 7D gel panel, lane 2**). Sequencing of the purified RT-PCR amplicons confirmed the MHS of 4 nts in the RNA junction of chimeric Krt10-MmuPV1 transcripts (**Fig 7D, right chromatogram**).

## Altered expression of the DNA repair factors H2ax, Fen1, Cdk1, Plk1, Polθ, Mdc1, and phosphorylation of H2AX and CtIP in MmuPV1 infection-induced tumors

Papillomavirus infection activates the host DNA damage response [51, 52]. Many DNA repair factors contribute to MMEJ in mammalian and yeast cells [25, 53–55]. To determine whether the expression of the host factors involved in MRN (MRE11-RAD50-NBS1)-CtIP-dependent MMEJ might be altered during MmuPV1 infection and papilloma induction, we analyzed two separate RNA-seq datasets [42, 56] to determine if there was an increase in the expression of DNA repair genes, particularly in the 21 genes involved in MMEJ [26] in MmuPV1-induced tumor tissues compared to the MmuPV1-infected tumor-free or normal control tissues (**S9 Table**). We found that there was, in MmuPV1-induced tumors, increased RNA levels for the DNA double-strand break (DSB) responder H2ax (1.7-fold) [57, 58], flap endonuclease 1 (Fen1, 1.6-fold) [59, 60], two mitotic kinases Cdk1 (1.5-fold) and Plk1 (1.3 -fold), and DNA polymerase theta (Polθ, 1.5-fold) [61–64], but there was a decrease in the expression of a CtIP inhibitor, mediator of DNA damage checkpoint 1 (Mdc1, -1.4-fold) [26] (**Fig 8A**). The increase in the levels of expression in these genes was greater in the MmuPV1-induced tumors than in the MmuPV1-infected tumor-free tissues (**S9 Table**). We verified the increased RNA expression of MMEJ-related H2ax, Fen1, Cdk1, Plk1, and Polθ and decreased RNA expression

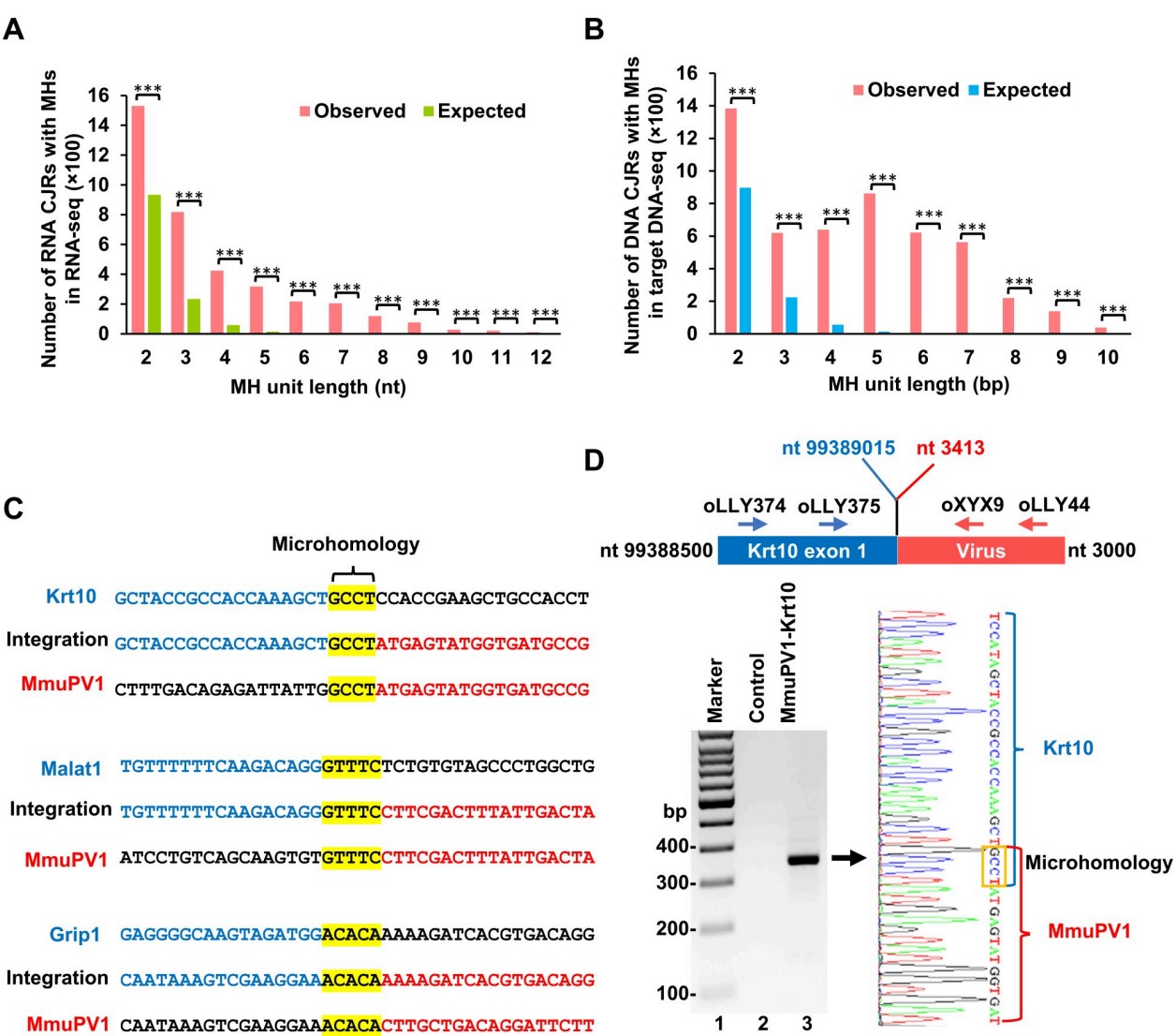

**Fig 7. Identification of microhomology sequences in the MmuPV1 integration junctions. A and B,** Microhomology sequences (MHS) are enriched in the junction regions of CJRs identified by RNA-seq (**A**) and targeted DNA-seq (**B**) when compared to the expected MHS at the hypothetic junction region of CJRs. ***, P<0.001 in chi-squared test. **C**, MHS identified at the MmuPV1-host integration junctions in Krt10, Malat1 and Grip1 by RNA-seq. Yellow color shows the MHS at the integration junction between MmuPV1 (NC_014326) and mouse (mm10) reference genomes. Blue and red colors indicate the upstream and downstream host and viral sequences at the integration site junction. **D,** Verification of MHS in a MmuPV1-Krt10 integration site from muzzle tumor tissues of animal #2 (**Fig 2B** for sample #6) by RT-PCR and sequencing. A pair of primers oLLY374 (located on Krt10) and oLLY44 (located on MmuPV1 E2 region) were used for the first round PCR and the primer pair of oLLY375 (located on Krt10) and oXYX9 (located on MmuPV1 E2 region) were used for nested PCR (top diagram). A primer pair, oMA1 from the Gapdh gene and oLLY44 from MmuPV1, was used as a negative control. See details in **S10 Table** for all primer sequences and locations. The MHS (GCCT) in the amplicons (gel panel, lane 3) was confirmed by Sanger sequencing (right) and highlighted in a yellow box.

of Mdc1 in the MmuPV1 tumor tissues by real-time RT-qPCR (**Fig 8B**). We also verified the increased phosphorylation of H2AX protein at S139 (also called γH2AX) and the increased expression of total FEN1 protein in the two MmuPV1 tumor tissues (**samples 1 and 4 in Fig 2B**) relative to the two tumor-free tissues (**samples 10 and 12 in Fig 2B**) (**Fig 8C**).

CtIP and FEN1 are important factors in the MMEJ-mediated DSB repair pathway that mediate DNA end-repair in ways that permit microhomology mediated end-joining [65–69]. Cdk1 and Plk1 are two protein kinases that are responsible for the phosphorylation and

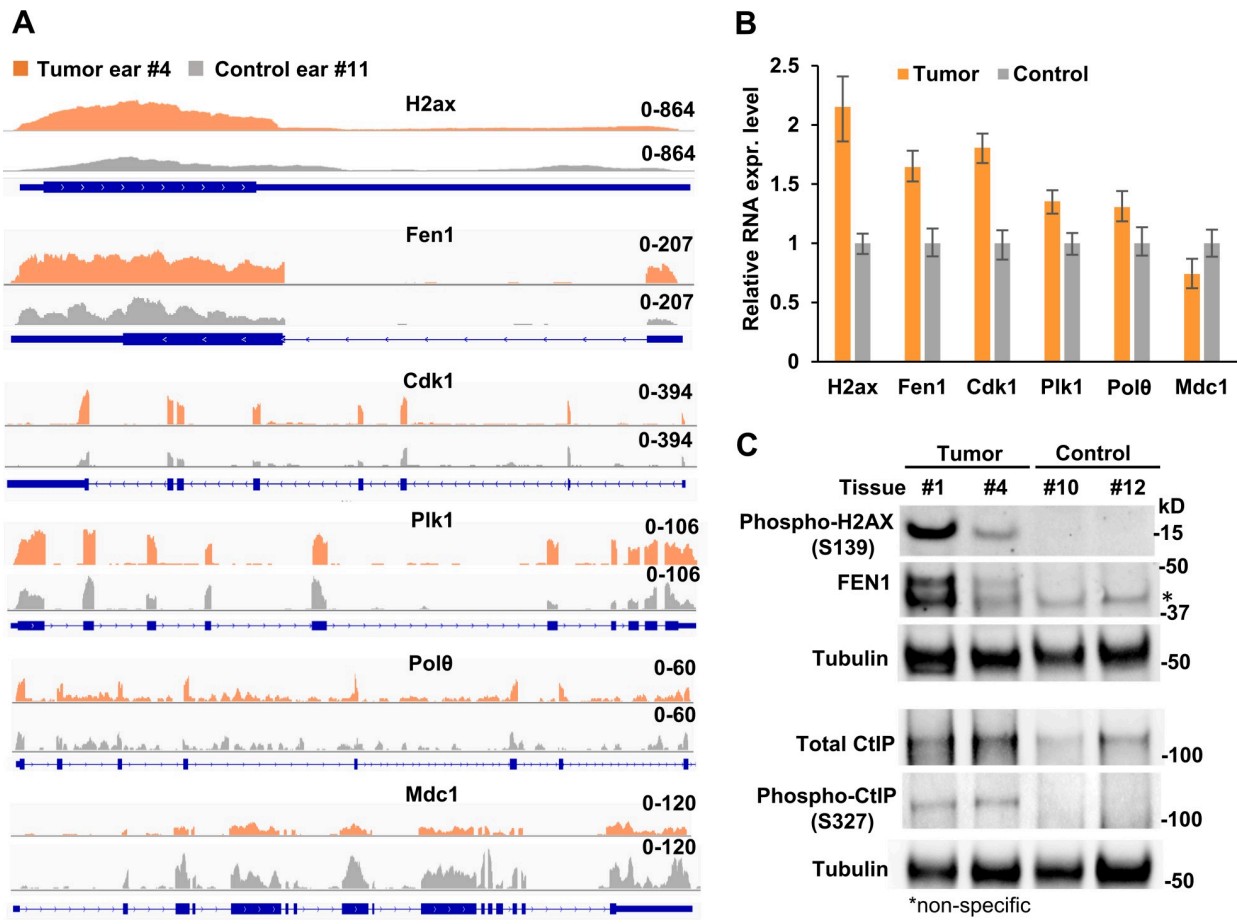

**Fig 8. Expression of host DNA repair factors in the MmuPV1-induced tumor and MmuPV1-infected, tumor-free tissues. A,** Distributions of mapped RNA-seq reads for H2ax, Fen1, Cdk1, Plk1, Polθ, and Mdc1 in tumor (Sample #4) and tumor-free (sample #11) ear tissues (**see Fig 2B**) to the mouse genome were visualized by IGV. **B,** Quantitative RNA expression for H2ax, Fen1, Cdk1, Plk1, Polθ, and Mdc1 from ear tissues by RT-qPCR. Total RNA extracted from MmuPV1 tumor (sample #4) and tumor-free (sample #11) ear tissues from animal #2 (**See Fig 2B**). The RNA expression levels of the indicated genes were determined by RT-qPCR using mouse gene-specific TaqMan probes. **C,** Phosphor-H2AX (S139) (γH2AX), FEN1, CtIP, and phosphor-CtIP (S327) in tumor samples #1 and #4 and tumor-free samples #10 and #12) (**see Fig 2B**) were determined by Western blot using the indicated antibodies. Tubulin served as a sample loading control.

activation of CtIP [70, 71]. Thus, the increased Cdk1 and Plk1 expression in MmuPV1-induced tumor tissues may promote the expression and phosphorylation of CtIP. Immunoblot assays were performed for CtIP on two MmuPV1 ear tumor tissues (**samples 1 and 4 in Fig 2B**) and two MmuPV1-infected, tumor-free ear tissues (**samples 10 and 12 in Fig 2B**). There was an increase in the amount of CtIP protein in the two MmuPV1 tumor tissues relative to the two tumor-free tissues (**Fig 8C**). CtIP that was phosphorylated at S327 was found only in two MmuPV1 ear tumors, but not in two tumor-free ear tissues (**Fig 8C**). These data suggest that the increased expression of Cdk1 and Plk1 in the tumor ears could be responsible for the increased phosphorylation of CtIP [70].

## Discussion

The MmuPV1-infected mouse can be used as a model of human papillomavirus pathogenesis [56, 72, 73]. Cutaneous, mucosal and blood infections of mice with MmuPV1 lead to the

development of benign papillomas (warts) and to high-grade precancerous lesions and carcinomas [32, 73, 74]. We found that MmuPV1 integration is common in MmuPV1-induced tumors in the infected mouse ear, tail, and muzzle. Like integrated HPV DNA [12, 13], integrated MmuPV1 were often linearized in the region of the E2 gene (**Fig 2C**). This is likely to be the result of post-integration selection on the growth of the cells and not preferential integration in this region of the viral genome. Linearization of the MmuPV1 genome in E2 prevents the expression of the normal MmuPV1 early transcripts by displacing a poly-A signal that is used for viral early RNAs.

We found that MmuPV1 DNA could be inserted in many places in the host genome, although we did not determine what fraction of cells in the tumor have integrated viral DNA. The majority of the host genome is either intergenic or intronic, and, as expected, we found MmuPV1 DNA was frequently integrated in the intergenic and intronic regions. There are also integrations in repeat sequences. Only a small fraction (4%) of the integrated MmuPV1 DNA was found in coding exons; however, coding exons comprise only ~2.6% of the mouse genome. Although MmuPV1 DNA was frequently integrated in intergenic regions, only a small fraction of the MmuPV1 DNA that was integrated in intergenic regions was expressed (**Figs 2F and 4D).** This is probably due to fact that there are no host promoters in the intergenic regions. For the integrated MmuPV1 DNAs that were linearized in the E2 region, expression of the viral sequences would require a poly-A signal be provided by the adjacent host sequences [75].

Using RNA-seq and RACE-SMRT-seq, we have identified multiple MmuPV1 integration sites in a group of genes whose lengths ranged from 0.3 kb to 87.6 kb (**Fig 3B**). These genes include both highly expressed and poorly expressed genes. It was difficult to determine the relative levels of expression for these genes by RNA-seq analysis in the MmuPV1 tumor and non-tumor tissues because the tissues were intermingled. However, by using RNA ISH (RNAscope technology), we showed that there was reduced expression of Krt10 and Fabp5 in MmuPV1 tumor tissues. We think that the decreased expression of these two genes was a consequence of the virus inhibiting differentiation, a hallmark of papillomavirus-induced lesions, and was not directly related to the places MmuPV1 DNA was integrated.

MMEJ is a Ku-independent, Polθ-dependent, DNA DSB repair pathway [26, 62–64] that could be involved aberrant IN-independent HIV-1 integration [47–49] and adeno-associated virus, papillomavirus and MCV integration[7, 13, 28, 29]. We showed that a majority of the virus-host junctions bear 2–10 nts of MHS between MmuPV1 and host DNA. By RT-PCR, we also identified these MHS in the MmuPV1-Krt10 RNA transcripts in a MmuPV1 tumor tissue. Our data show that, in this respect, MmuPV1 and HPV integrations are similar [7, 13].

Five steps (end resection, annealing of the MHS, flap removal, fill-in synthesis and ligation) and more than ten host factors are involved in MMEJ. The first step for MMEJ is a limited resection of the ends of the DSB by host nucleases, which exposes the microhomologous sequences [25, 54]. The host nucleases CtIP and FEN1 are two of the factors that generate a 3' overhang, setting the stage for Polθ-mediated MMEJ [62, 63, 68, 71, 76–78]. To be active, CtIP needs to be phosphorylated at S327, S723 and T859 by three host serine/threonine protein kinases: CDK1, Aurora A, and PLK1 [67, 68, 70]. We found that MmuPV1 tumors displayed the increased levels of both Fen1 RNA and protein and had higher levels of both Cdk1 and Plk1 RNA when compared to MmuPV1-infected tissues that had not become tumors. Increased CtIP expression and S327 phosphorylation were found only in MmuPV1 tumor tissues, not in MmuPV1-infected tumor-free tissues. A recent report indicated that CtIP phosphorylation at S327 by CDK1/Aurora A triggers CtIP binding to PLK1 which facilitates PLK1 phosphorylation of CtIP at S723, thereby initiating the end resection [70].

The initial MmuPV1 DNA integrations must take place in infected but still normal tissues. Thus, the increased expression of the MMEJ host factors would not have played a role in the initiating events. However, the amount of integrated viral DNA is higher in the tumor tissues and there are step-wise increases in the expression of these host genes from normal tissues, to tumor-free MmuPV1 infected tissues, and then to MmuPV1 tumors (**S9 Table**). This increase might play a role in the increased levels of viral DNA integration in tumors (**Fig 5**).

Whether MmuPV1 infection contributes to the DNA break/damage of the viral and host genomes that leads to viral DNA integration remains to be investigated. We found that increased expression of γH2AX, a sensitive molecular marker of DNA damage [57, 58], was detected only in MmuPV1-induced tumor, but not from MmuPV1-infected, tumor-free tissues. If viral factors are involved, the roles they play are not defined. Integration of viral DNA would be facilitated by the very high levels of viral DNA that are present in the infected cells with increased expression of DNA repair factors. Papillomavirus E7 in cell cultures is known to activate the expression of both ATM-and ATR-mediated DNA damage genes whose expression is critical for viral DNA replication [51, 79]. E7 in normal human foreskin keratinocytes induces mitotic defects and host genome instability with increased γH2AX expression [20–22], presumably by promoting host DNA replication and increased both homologous recombination and MMEJ of a GFP reporter DNA with DSBs in U2OS cells [80], an osteosarcoma cell line that does not support papillomavirus infection.

Here we propose a model of how MmuPV1infection may lead to viral integration and tumorigenesis (**Fig 9**). In this model, MmuPV1 infection of host keratinocytes induces a DNA damage response that is important for viral DNA replication and also triggers instability of both viral and host genome, probably through the actions of E7 and/or other viral and host proteins. If the viral genome is linearized, it could be accidently inserted into any nicked or damaged region of the host genome, most commonly by the MMEJ-mediated DNA repair pathway. Although viral DNA integration is a rare event during virus infection and a dead end for productive MmuPV1 infection, integrated copies of MmuPV1 DNA accumulate in the host genome as a function of time following MmuPV1 infection. The increase in integrated viral DNA is accompanied by increased expression of H2ax, Fen1, Cdk1, Plk1, and Polθ and CtIP phosphorylation. This increase in the amount of integrated MmuPV1 DNA promotes tumor formation. High levels of integrated viral DNA through the linearized E2 region causes an increase in the expression of the viral oncogenes [14–16] which leads to clonal expansion of the cells, tumor progression, and ultimately to cancer.

## Materials and methods

### Ethic statements

All procedures that involved animals and their maintenance were conducted in accordance with protocols approved by the University of Wisconsin Medical School Institutional Animal Care and Use Committee (IACUC, Protocol number: M02478). Mice were housed at McArdle Laboratory Animal Care Unit in strict accordance with guidelines approved by the Association for Assessment of Laboratory Animal Care (AALAC), at the University of Wisconsin Medical School.

### MmuPV1 mouse model and sample collection

A description of the MmuPV1 mouse model was previously reported [74]. Briefly, three immunodeficient athymic BALB/c FoxN1$^{nu/nu}$ female mice (6–8 weeks old) were infected with a fixed amount of MmuPV1 ($10^8$ viral genome equivalents (VGE) per infection site) after scarifying skin of the tail, ear, and muzzle. The wart tissues from the three anatomical sites (ear,

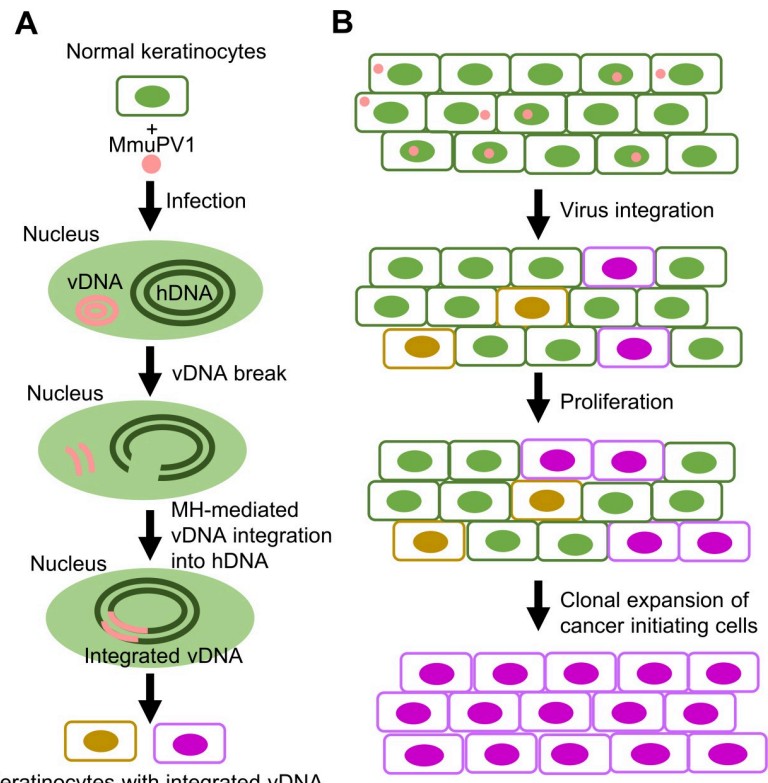

**Fig 9. Proposed model of the role of integrated MmuPV1 DNA in tumorigenesis.** After virus infection, MmuPV1 induction of the host DNA damage response and activation of the cellular DNA repair machinery enhance the breakage and recombination of both the viral and host genomes. The linearized viral genomes (vDNA) could be fused to any nicked or damaged region of the host genome (hDNA) most likely by a microhomology-mediated DNA repair pathway (**A**). MmuPV1 integration into the host genomic DNA appears to be mostly random; however, what we detected in our experiments was modified by post-integration selection. Integrated MmuPV1 DNA accumulates over time following MmuPV1 infection, contributing to tumor formation. Clonal expansion of cells that express viral E6 and E7 from the integrated MmuPV1 DNA through the linearized E2 region [14–16], leads to tumor progression and ultimately to cancer (**B**).

tail and muzzle) were collected from each animal after six months or at a specified time post-infection. The samples were snap-frozen in liquid nitrogen for RNA/DNA/protein isolation. Papilloma tissues were also excised, fixed in 10% neutral buffered formalin and embedded in paraffin. Serial sections (5 μm thick) were prepared for H&E staining and subsequent analyses.

## Total RNA sequencing and bioinformatics

RNAs extracted from mouse tissues using the TriPure reagent (Roche, #11667165001) were treated with the TURBO DNA-free Kit (Thermo Fisher Scientific, #AM1907) to eliminate any residual of DNA. cDNA libraries were constructed following the Illumina Stranded Total RNA protocol (Illumina, #RS-122-2201) and sequenced on Illumina HiSeq 2500 with 2×125 nts modality and a depth of 100 million reads per sample. The reads were trimmed using cutadapt v 1.16 (DOI:10.14806/ej.17.1.200) to filter out the low quality reads and remove adaptor sequences. Quality control was performed using the FastQC (http://www.bioinformatics.babraham.ac.uk) package and reads were checked for contamination with FastqScreen v 0.9.3 (https://doi.org/10.12688/f1000research.15931.2). cDNA sequences were matched to the sequences in MmuPV1 (NC_014326) and the mouse genome (GRCm38-mm10) and the reads

were aligned using STAR v 2.5.2b aligner package [81]. Paired-end reads are two reads derived from the same cDNA fragment with one read from the adapter ligated to the end of the fragment that corresponds to the 5' end of the RNA and the other from the adapter ligated to the end of the fragment that corresponds to 3' end of the RNA. The paired-end reads that perfectly aligned with only the MmuPV1 or the mouse genome were removed, and the remaining "chimeric reads" were analyzed. We identified two types of chimeric reads: chimeric paired reads and chimeric junction reads. The chimeric paired reads are the paired-end reads in which one read is fully aligned to MmuPV1 genome (as defined by aligner), while the other read is perfectly aligned to mouse genome. This suggests that there is an integration junction between the two reads. A chimeric junction read (CJR) is a read that is partially aligned to the both MmuPV1 and mouse genomes for at least 20 nts. Although a small fraction of the host/virus RNA junctions derive from splicing events, we have excluded those in which the junctions were created by splicing from the data we present (see Results). Microhomology calculations were based on the number of overlapped nucleotides identified by STAR between virus and host sequences at the junctions of chimeric reads (the integration site junction reads). Raw data and the analyzed RNA-seq data supporting the findings in this study have been deposited in the NCBI GEO database (the accession #: GSE163537).

## Rapid amplification of cDNA ends (RACE) and single molecule real-time sequencing (SMRT-seq)

The Smart RACE cDNA amplification kit (Clontech, #634858) was used, following the manufacturer's instructions, to perform 5' /3' RACE assays with 1 μg/reaction of total RNA as the template. The primers used in the assays are listed in **S10 Table**. The 5' RACE products were sequenced by SMRT-seq using PacBio Iso-seq technology (Pacific Biosciences). The 3' RACE products were size selected using BluePippin (Sage science, MA) to remove products smaller than 1500-bp. The details of SMRT-seq were described previously [42].

PacBio SMRT-seq Analysis Package (smrtpipe.py v2.0, with default settings) was used to process the sequence data into circular consensus sequences (CCS) that were used for further analyses. The CCS reads were aligned to MmuPV1 genome (NC_014326) and mouse genome (GRCm38-mm10) by STAR-long *(STARlong—runMode alignReads \—readNameSeparator space \—outFilterMultimapScoreRange 1 \—outFilterMismatchNmax 2000 \—winAnchorMultimapNmax 200 \—coreGapNoncan -20 \—scoreGapGCAG -4 \—scoreGapATAC -8 \—scoreDelBase -1 \—scoreDelOpen -1 \—scoreInsOpen -1 \—scoreInsBase -1 \—seedSearchLmax 30 \—seedSearchStartLmax 50 \—seedPerReadNmax 100000 \—seedPerWindowNmax 1000 \—alignTranscriptsPerReadNmax 100000 \—alignTranscriptsPerWindowNmax 10000 \—alignEndsType Local)* after removing the adapter sequences, poly (A) tails, and the artificial chimeras. The clustering step was then performed using hierarchical n*log(n) alignment and iterative cluster merging to cluster similar isoforms together. The chimeric transcripts are defined as CCS reads with at least 50 bp aligned to the mouse genome and at least 100 bp aligned to MmuPV1 genome.

## Mouse genomic DNA extraction and droplet digital PCR (ddPCR)

Mouse tissues were digested with proteinase K at 50˚C overnight and the genomic DNA was precipitated with ethanol in the presence of sodium chloride. The DNA concentration was measured by Nanodrop ND-1000 spectrophotometer (Thermo Fisher Scientific).

The number of MmuPV1 DNA copies per cell in the infected mouse tissues was determined by ddPCR. Three genomic DNA samples were prepared from MmuPV1-infected mouse tissues and one control genomic DNA sample was prepared from a normal mouse tissue free

from MmuPV1 infection. All of the DNA samples were diluted each to 10 ng/µl with a low-EDTA TE buffer (10 mM Tris-HCl, 0.1 mM EDTA, pH = 8.0). The mouse transferrin receptor Tfrc gene was used as an internal reference gene to determine the input cell number that was used for viral DNA copy number determinations. Briefly, mouse genomic DNAs from MmuPV1-infected tissues were diluted 1:500 in control DNA. Each sample of approximately 10 ng of diluted genomic DNA was used, in duplicate 20-µl ddPCR reactions containing the primers and probes for the MmuPV1 (**S10 Table**) and Tfrc (Thermo Fisher Scientific, #4458366). Droplet generation (BioRad QX200) and PCR (BioRad T100) were performed according to the manufacturer's protocol (BioRad). The cycling protocol started with a 95˚C enzyme activation step for 10 min and followed by 40 cycles of a two-step cycling (94˚C for 30 seconds and 60˚C for 30 seconds at ramping rate of 2˚C/sec). The final extension time was 10 min at 98˚C. BioRad QuantaSoft 1.5.38.1118 was used to process the data. The cell number in the samples was estimated as one-half of the Tfrc copy number because each cell has 2 copies of the Tfrc gene. After normalizing the data, the mean MmuPV1 DNA copy number per cell was calculated for each sample from the duplicate ddPCR reactions.

### Targeted DNA sequencing (DNA-seq)

One microgram of genomic DNA was fractionated by gel electrophoresis to remove the majority of the episomal viral genomic DNA (the size of the genome is ~8 kb). The host DNA was recovered and sheared into 300–500 bp long fragments using a Covaris M220 Focused-ultrasonicator (Covaris). The fragmented DNAs (~500 ng) were end repaired and dA-tailed using the NEBNext Ultra Ligation kit (NEB, #E7445) according to the manufacturer's instructions. A partially double stranded linker (5'-GTAATACGACTCACTATAGCGCTGCGCTTAAGCG ACT-3', 5'–PO$_4$-GTCGCTTAAGCGCAG-3'-C6) with a 3' T overhang was ligated to the genomic DNA fragments using the NEB Quick Ligation Kit (NEB, M2200). Virus-specific primers (Pr2976/Pr3277/Pr3742) were designed based on RNA-seq results and were used to selectively amplify the integration site junctions. Two rounds of 20 PCR cycles were conducted to enrich the integration junctions in the sequencing libraries using the Phusion High-Fidelity PCR Kit (NEB, #E0553). Paired-end sequencing using the 150-bp modality was performed on the HiSeq3000 sequencer according to the manufacturer's instructions (Illumina). Reads were preprocessed in a same way as described in "Total RNA sequencing" and aligned to the chimeric mouse-virus genome using STAR v 2.5.2b with the following parameters: "*—outFilterIntron-Motifs RemoveNoncanonicalUnannotated—outSAMstrandField None—outFilterType BySJout—outFilterMultimapNmax 20—alignSJoverhangMin 8—alignSJDBoverhangMin 1—outFilter-MismatchNmax 999—outFilterMismatchNoverLmax 0.3—alignIntronMin 20—alignIntron-Max 1000000—alignMatesGapMax 1000000—clip3pAdapterSeq -—readFilesIn R1.trim.fastq.gz R2.trim.fastq.gz—readFilesCommand zcat—runThreadN 32—outFileNamePrefix OutputName—outSAMtype BAM Unsorted—chimSegmentMin 20*". Once "chimeric junction reads" and "chimeric paired reads" were identified an additional alignment was performed using the default parameters to filter out poorly mapped reads. Raw data and the analyzed targeted DNA-seq data supporting the findings in this study have been deposited in the NCBI GEO database (accession #: GSE163681).

### Computational analyses of the MmuPV1 integration sites in DNA repeat regions and identification of microhomologous sequences (MHS)

To ask if there are MmuPV1 integration sites in repeat regions in the mouse genome we looked for MmuPV1 in the following repeat elements: LINEs, SINEs, LTR elements, satellite, and simple repeats, which respectively comprise 21.2%,13.4%, 8.7%, 2.6%, and 1.5% of the

mouse genome (http://www.repeatmasker.org/species/mm.html) [82]. The fractions served as a measure of the amounts of the DNA in each type of repeat which could be compared with the frequency of the MmuPV1 integrations observed in these repeat regions in the mouse genome. The significance of the data was determined using a chi-squared test. The significance of the data in multiple comparisons was adjusted using the Bonferroni correction.

The microhomologies at the virus-host chimeric junctions were determined using STAR v 2.5.2b. The expected frequencies at which MHS would arise at the junctions was calculated as $0.25^{(\text{length of MH unit})}$. The significance of the data was determined using a chi-squared test and the significance of the data in multiple comparisons was adjusted using the Bonferroni correction.

## RNAscope *in situ* hybridization (ISH)

The *in situ* detection of viral and host transcripts was performed as described [42]. Briefly, MmuPV1-induced tumor tissues were fixed using 10% formalin in PBS (pH 6.8–7.2) for 20 h at room temperature, dehydrated, and embedded in paraffin. The tissues were cut into 5 μm sections using a microtome. RNA *in situ* hybridization (RNA-ISH) was performed using RNA-scope technology (Advanced Cell Diagnostics or ACD, Newark, CA) with RNAscope 2.5 HD detection reagents—BROWN (ACD, # 322310) for single staining or RNAscope Multiplex Fluorescent Detection Reagents V2 (ACD, # 323110) for florescence dual staining as recommended by the manufacturer. To remove the signal from viral genomic DNA, all tissue sections were rehydrated, digested first with RNAscope Protease Plus for 30 min, and then with 20 units of DNase I (Thermo Fisher Scientific, #EN0521, diluted in 1 × reaction buffer with $MgCl_2$) for 30 min at 40°C. The single-plex RNAscope assay uses an antisense probe to detect the specific RNA; the standard probes are referred to as channel 1 probes. MmuPV1 L1 probe (ACD, #473278) for channel 1 was used for single (brown) staining. For dual staining, one of the probes can be in channel 1, but the second probe must be specifically designed by ACD for use in the dual staining approach (a "channel 2" probe). In the current study, a MmuPV1 E6E7 (ACD, #409771-C2) probe was created for channel 2 staining and was mixed at 1:50 dilution with a mouse Krt10 (ACD, #457901) or Fabp5 (ACD, #504331) probe for channel 1 before hybridization. The signal was detected using the TSA Fluorescein Evaluation Kit (Perkin Elmer, # NEL760001KT) and scanned at 20× resolution using an Aperio CS2 Digital Pathology Scanner (Leica Biosystem). The intensities of the hybridized RNA signals in the tumor tissue, relative to normal tissue surrounding the tumor, were quantified by using ImageJ software (https://imagej.nih.gov/ij/).

## Cell culture and siRNA knockdown

Primary mouse keratinocytes from newborn C57Bl/6NCr mouse were cultivated as described [83] in the presence of Rho kinase inhibitor Y-27632 (Enzo Life Sciences, #ALX-270-333). SMARTpool human siRNAs targeting Pard3 (#M-040036-01-0005) and Grip1(#M-064716-00-0005) were purchased from Dharmacon. Non-targeting control siRNA (Dharmacon, # D-001210-01) served as a non-specific siRNA negative control. Mouse keratinocytes plated without the Rho kinase inhibitor Y-27632 [84] were transfected twice with each siRNA (40 nM) at an interval of 24 h using the LipoJet In Vitro Transfection Kit (Ver. II) (SignaGen Laboratories, #SL100468). Total protein and total RNA extracts were prepared 24 h after the second siRNA transfection.

## Cell proliferation and flow cytometry

For the cell proliferation assays, the dehydrogenase activities in living cells were measured using the Cell Counting Kit-8 (CCK-8) from Dojindo Molecular Technologies. Briefly, mouse

primary keratinocytes were incubated with 10% WST (water-soluble tetrazolium salt)-8 in culture medium for 1 h at 37˚C, and the absorbance at 450 nm of the cell culture medium (measured in six repeats) was used to calculate the relative cell viability.

For the cell cycle distribution analysis, 1.5 million cells were fixed with 70% ethanol at 4˚C overnight, washed with PBS, and then incubated in PBS containing 200 ug/ml RNase A (Thermo Fisher Scientific, #P3566), 20 µg/ml propidium iodide (Thermo Fisher Scientific, #EN0531), and 0.1% Triton X-100 (Promega, #H5142) at room temperature for 30 min. Cell cycle distribution was determined by flow cytometry using LSRFortessa (BD Biosciences). The data were analyzed by FlowJo software (FlowJo).

## RT-PCR and RT-qPCR

Total RNA was extracted using the TriPure reagent (Roche, #11667165001), treated with DNase TURBO (Thermo Fisher Scientific, #AM1907), and converted to cDNA using MuLV reverse transcriptase and random hexamers. PCR amplification was performed with AmpliTaq (Thermo Fisher Scientific) using the primers listed in **S10 Table**. RT-qPCR was carried out using TaqMan Gene Expression Master Mix (Thermo Fisher Scientific, #4369016) with gene-specific probes of Cdk1(#Mm00772472_m1), Plk1(#Mm00440924_m1), Fen1 (#Mm01700195_m1), Polθ (#Mm01170059_m1), Mdc1(#Mm01273851_g1), or H2ax (Mm00515990_s1).

## Western blot and antibodies

For primary mouse keratinocytes, total protein was prepared from 2 million cells by directly lysing the cells in 200 µl of 2× SDS protein sample buffer containing 5% 2- mercaptoethanol (2-ME). Mouse tissues were dissected and homogenized in RIPA buffer (500 µl/10 mg tissues; Boston Bio-Products, #BP-115X) with the addition of cocktail of protease inhibitors for chymotrypsin, thermolysin, papain, pronase, pancreatic extract, trypsin (Roche, #04693159001) and phosphatase inhibitors (sodium fluoride, sodium orthovanadate, sodium pyrophosphate and β-glycerophosphate) (Thermo Fisher Scientific, #78420). 24 µl of the lysate was then mixed with 6 µl 4× SDS-loading buffer with addition of 2-ME, heat denatured at 95˚C for 10 minutes, and loaded onto a NuPAGE Bis-Tris 4–12% gel (Thermo Fisher Scientific). After electrophoresis in 1× MOPS SDS buffer (Thermo Fisher Scientific, # NP0001), the fractionated proteins were transferred onto a nitro-cellulose membrane and blocked for 1 h with 5% skimmed milk in 1× TBS (Tris-buffered saline) containing 0.05% Tween-20 (TTBS). The primary antibody diluted in TTBS and incubated with the membrane overnight at 4˚C followed by 1 h at room temperature incubation. After washing the membrane three times with 1× TTBS buffer, the membranes were then incubated with an appropriate peroxidase-conjugated secondary antibody diluted in 2% milk/TTBS. After extensive washes with 1× TTBS, the bound antibody on the membrane was detected with SuperSignal West Pico (Thermo Fisher Scientific, # 34080), visualized and captured by an ImageLab system (Bio-Rad).

The following antibodies were used for Western blotting: anti-Pard3 (Abcam, #ab191204), anti-Grip1(Abcam, #ab226201), anti-GAPDH (Cell Signaling, #14C10), anti-beta tubulin (Sigma-Aldrich, #T5201), anti-CtIP (Cell Signaling, #D76F7), anti-phospho-CtIP (Thermo Fisher Scientific, #PA5-37337), anti-FEN1 (Cell Signaling, #82354), and anti-phospho-H2AX (Cell Signaling, #9718) were used as primary antibodies. Goat anti-rabbit IgG (whole molecule)–peroxidase antibody (Sigma-Aldrich, #A0545) and goat anti-mouse IgG (Fc specific)–peroxidase antibody (Sigma-Aldrich, #A2554) were used as secondary antibodies.

## Supporting information

**S1 Fig. SMRT-seq analyses of MmuPV1 integration sites in mouse tumor tissues. A and B,** 5′ RACE products (A) and 3′ RACE products (B) of MmuPV1 transcripts were identified

using different MmuPV1-specific primers on total RNA isolated from MmuPV1-induced tumor tissues. All the products from 5′ RACE were analyzed by SMRT-seq to identify chimeric virus-host RNA transcripts (A). After removing the major viral polyadenylation products derived from transcription of MmuPV1 episomal DNA (red rectangles), the remaining 3′ RACE products (B) were analyzed by SMRT-seq for chimeric virus-host RNA transcripts. **C and D,** Distribution of integration breakpoints across the MmuPV1 genome identified by 5′ RACE (C)- and 3′ RACE (D)-SMRT-seq. **E and F,** Top10 host genes with MmuPV1 integrated DNA as detected by 5′ RACE (E)- and 3′ RACE (F)-SMRT-seq.
(TIF)

**S2 Fig. Distribution of DNA CJRs in the interspersed repeats in the mouse genome. A,** DNA CJRs from MmuPV1 tumor and non-tumor (control) tissues were identified by targeted DNA-seq. MmuPV1 copy number per cell was determined by ddPCR from 10 ng of genomic DNA using the mouse Tfrc gene as an internal control. **B,** Expected and observed CJRs in different interspersed repeats in the mouse genome identified by targeted DNA-seq. Repeat data for the mm10 genome was downloaded from repeatmasker.org website (Repeat Library 20140131). "intersectBed" command from bedtools (https://doi.org/10.1093/bioinformatics/btq033) package was used to identify the virus-host junctions that mapped to known repeat regions. **, $P<0.01$ by a chi-squared test.
(TIF)

**S3 Fig. Possible functions for host genes in the response to MmuPV1 infection, integration, and tumorigenesis. A,** Flow cytometry analysis of mouse keratinocytes with reduced expression of Pard3 and Grip1 after gene-specific siRNA treatment. The mouse primary keratinocytes were transfected twice with 40 nM of siRNA (si-Pard3, Si-Grip1 or si-NS) at a 24 h interval. Cells were fixed 24 h after the second siRNA knockdown (KD) and analyzed by flow cytometry in triplicate. PI, propidium iodide. **B,** Bar graphs show the cell cycle distribution after two rounds of siRNA KD of Pard3 and Grip1 expression. Data are the mean ± SD (n = 3). * $P<0.05$ by paired, two-tailed Student's *t* test.
(TIF)

**S1 Table. Top 100 genes with MmuPV1 integration sites identified by RNA-seq analyses.**
(XLSX)

**S2 Table. Mapping of 5' RACE-SMRT-seq clustered reads.**
(XLSX)

**S3 Table. Mapping of 3' RACE-SMRT-seq clustered reads.**
(XLSX)

**S4 Table. 231 mouse genes with MmuPV1 integration sites identified by both RNA-seq and Targeted DNA-seq.**
(XLSX)

**S5 Table. Top 100 genes with MmuPV1 integration sites identified by targeted DNA-seq.**
(XLSX)

**S6 Table. Genes with MmuPV1 integration sites identified by targeted DNA-seq in MmuPV1-induced tumors of the ear and muzzle and MmuPV1-infected tumor-free ear tissues.**
(XLSX)

**S7 Table. Differentially expressed genes in three ear tumor samples (S1/S4/S7) and three non-tumor ear samples (S10/S11/S12).**
(XLSX)

**S8 Table. Relative RNA expression of 40 tumor-specific genes with MmuPV1 integration sites identified by targeted DNA-seq in ear tissues.** Data were from RNA-seq analysis.
(XLSX)

**S9 Table. Altered expression of host genes involved in MMEJ in MmuPV1 tumor and non-tumor tissues identified by two separate RNA-seq analyses.**
(XLSX)

**S10 Table. DNA oligoes used in the study.**
(XLSX)

## Acknowledgments

We thank Alison McBride of NIAID for providing mouse keratinocytes for this work. We thank Binghui Shen at City of Hope National Medical Center for his critical comments on host factors in MMEJ, Baktiar Karim of Frederick National Laboratory for Cancer Research for his pathology expertise, Xiaolin Wu and David Sun of Frederick National Laboratory for Cancer Research for their ddPCR analysis.

## Author Contributions

**Conceptualization:** Lulu Yu, Paul F. Lambert, Zhi-Ming Zheng.

**Data curation:** Lulu Yu, Vladimir Majerciak, Xiang-Yang Xue, Aayushi Uberoi.

**Formal analysis:** Lulu Yu, Vladimir Majerciak, Xiang-Yang Xue, Aayushi Uberoi, Alexei Lobanov, Xiongfong Chen, Maggie Cam, Stephen H. Hughes, Paul F. Lambert, Zhi-Ming Zheng.

**Funding acquisition:** Paul F. Lambert, Zhi-Ming Zheng.

**Investigation:** Lulu Yu, Vladimir Majerciak, Xiang-Yang Xue, Aayushi Uberoi, Stephen H. Hughes, Paul F. Lambert, Zhi-Ming Zheng.

**Methodology:** Lulu Yu, Vladimir Majerciak, Xiang-Yang Xue, Aayushi Uberoi, Alexei Lobanov, Xiongfong Chen, Maggie Cam, Stephen H. Hughes, Paul F. Lambert, Zhi-Ming Zheng.

**Project administration:** Paul F. Lambert, Zhi-Ming Zheng.

**Resources:** Paul F. Lambert, Zhi-Ming Zheng.

**Software:** Alexei Lobanov, Xiongfong Chen, Maggie Cam.

**Supervision:** Stephen H. Hughes, Paul F. Lambert, Zhi-Ming Zheng.

**Validation:** Lulu Yu, Vladimir Majerciak, Xiang-Yang Xue, Aayushi Uberoi.

**Visualization:** Lulu Yu, Vladimir Majerciak, Aayushi Uberoi, Alexei Lobanov, Xiongfong Chen, Zhi-Ming Zheng.

**Writing – original draft:** Lulu Yu, Vladimir Majerciak, Zhi-Ming Zheng.

**Writing – review & editing:** Lulu Yu, Vladimir Majerciak, Aayushi Uberoi, Alexei Lobanov, Stephen H. Hughes, Paul F. Lambert, Zhi-Ming Zheng.

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
