## [Decision Letter · Decision Letter 0]

17 Jun 2021

Dear Dr. Zheng,

Thank you very much for submitting your manuscript "Mouse papillomavirus type 1 (MmuPV1) DNA is frequently integrated in benign tumors by microhomology-mediated end-joining" for consideration at PLOS Pathogens. As with all papers reviewed by the journal, your manuscript was reviewed by members of the editorial board and by several independent reviewers. The reviewers appreciated the attention to an important topic. Based on the reviews, we are likely to accept this manuscript for publication, providing that you modify the manuscript according to the review recommendations.

Sincerely,

Richard BS Roden

Associate Editor

PLOS Pathogens

Alison McBride

Section Editor

PLOS Pathogens

Kasturi Haldar

Editor-in-Chief

PLOS Pathogens

orcid.org/0000-0001-5065-158X

Michael Malim

Editor-in-Chief

PLOS Pathogens

orcid.org/0000-0002-7699-2064

Reviewer Comments (if any, and for reference):

Reviewer's Responses to Questions

**Part I - Summary**

Reviewer #1: Manuscript: PPATHOGENS-D-21-00964 Title: Mouse papillomavirus type 1 (MmuPV1) DNA is frequently integrated in benign tumors

by microhomology-mediated end-joining.

Authors: Lulu Yu, Vladimir Majerciak, Xiang-Yang Xue, Aayushi Uberoi, Alexei Lobanov, Xiongfong Chen, Maggie Cam, Stephen H. Hughes, Paul F. Lambert, and Zhi-Ming

Zheng.

This manuscript describes novel findings in the MmuPV1 mouse infection model to determine whether this virus infection can lead to viral integration that would lead to malignant progression, and further delineate additional mechanisms to explain the observations that MmuPV-1 infections can progress to malignancy. At present, the MmuPV1 model has shown clear evidence for malignant progression, but much of the current literature has focused on early infections and a mechanism for the malignant phenotype is lacking. The current study provides a comprehensive and mechanistic model for the potential for viral integration by demonstrating that even in the early stages of benign infections, there is evidence of viral transcripts that are spliced to host genetic elements. These events would increase the statistical likelihood that viral integration is possible in this model and that MmuPV1 could be categorized as a member of the PVs that demonstrate malignant potential in the natural host.

The study is a powerful and comprehensive documentation of these rare but potentially clinically significant integration events and set the stage for more in-depth analyses of HPV infections from more benign HPV types.

Some items for consideration:

1. The Abstract has several abbreviations that would benefit from expansion (or a list of abbreviations if not already in place).

2. The observation that integration events are observed early in infection is particularly novel as histological and current virological assays would identify MmuPV1 early infections as benign and that most viral DNA are episomal. Intriguingly, these studies also indicate that such relatively rare integration events are ongoing during the disease progression and are a novel explanation as to how HPV integration can lead to cancer (increased survival advantage of rare cells with integrated events). More importantly, such events may be ongoing in infections with other HPV types that are less carcinogenic, and this could also set the stage for cancer development due to the ongoing host-cell “insults”. These findings are worthy of a future in depth study of HPV lesions from more benign HPV types (particularly the various skin HPV types) that are suggested to be a co-factor in non-melanoma skin cancer despite the difficulty of detecting integrated HPV DNA in tumor tissues.

Reviewer #2: MmuPV1 serves as a valuable mouse model for papillomavirus infection in both immunocompromised and immunocompetent animals. This paper reports integration sites/frequency after experimental MmuPV1 infection of BALB/c FoxN1nu/nu mice (immunodeficient). Mice were typically infected for 6M to allow wart formation. Using RNA-seq, RACE-SMRT-seq, and/or targeted DNA-seq they describe integration, consistent with a Microhomology-mediated end-joining (MMEJ) mechanism. They provide interesting correlative data regarding increased integration and infection/neoplasia and host MMEJ factors that are increased following infection/integration, and speculate (with good discussion) regarding the mechanisms behind the observed phenomena. Integration often resulted in disruption of E2, which would be expected to cause a selective advantage of those integrants if they integrate near a usable host polyadenylation signal to enable expression of early viral genes.

The science is sound and the paper is well written and clear. The work is important, laying the ground for testable hypothesis moving forward, and very appropriate for this journal. A few concerns/issues are outlined below, which if addressed would strengthen the paper in my opinion.

Concerns/Issues:

1. The data in Figure 7. While the data show modest upregulation of these factors at the transcriptional level, their effects would be exerted at the protein level. Validation of this data by western blot would strengthen the conclusions from this data.

2. Figure 2b- the table indicates that there were ~1.3M viral reads for the mouse 1 infected ear, and ~972K viral reads for the control ear (uninfected) from the same animal. This seems like way too many viral reads from an uninfected control ear, can this be explained? Was there transmission from primary infections to these "uninfected" sites during those 6M? If so why weren't there any warts in these "uninfected" sites? (or were there?). Did any warts appear in uninfected sites during the experimental infections?

3. Line 389-390: "There were multiple integrations in Malat1, Flg, Krt1, Dsp, Krt10, Hmr and Rn7sk (Fig. 2E, Supplementary Table S2)." Are these the only genes with multiple integrations from this data? Table S2 seems to suggest many genes had multiple integrations, unless I'm confused?

4. Line 462-463: "although the DNA CJRs from these genes did not have the same junction sequences

that were identified in the RNA CRJs." Why is this? Additional discussion about these discrepancies is needed.

5. Similar to #2 above, RNA-seq library prep can lead to sequencing artifacts (as recently highlighted witht he whole SARS-CoV-2 integration debacle, and papers like PMID 33980601 refuting that shoddy work). This work would be improved with a thorough discussion of these technical issues, and what was done here to prevent or recognize artifactual sequencing data.

6. Figure 1A- please indicate/confirm this was a 6m infection in the text, figure and/or legend. It is unclear from reading the results text on line 352.

7. Are homologous sequences in the range of 2-3 nt really long enough to be classified as "microhomology"? These seem too short. Perhaps a comment about this or some citation of literature discussing this would improve. If the 2/3 nt data are removed does it affect the conclusions?

Reviewer #3: Mouse papillomavirus type 1 (MmuPV1) DNA is frequently integrated in benign tumors

by microhomology-mediated end-joining

The authors investigated integration of the murine papillomavirus MmuPV1 into the genome of

BALB/c FoxN1nu/nu mice at three skin sites known to be susceptible to MmuPV1-induced papilloma formation (tail, muzzle, ear). They provide evidence that the viral DNA is integrated in infected cells, particularly in benign skin papillomas. This topic is of utmost interest to the papillomavirus field and the findings presented herein are novel. The manuscript is well written, contain important data and the experiments are sound and well performed.

**Part II – Major Issues: Key Experiments Required for Acceptance**

Reviewer #1: None

Reviewer #2: see above- western blots to validate findings in Figure 7. Additional analysis if 2nt or 3nt "microhomology" data are removed (unless these are indeed considered valid nt lengths for MMEJ mechanisms)

Reviewer #3: none required

**Part III – Minor Issues: Editorial and Data Presentation Modifications**

Reviewer #1: See above

Reviewer #2: see above, point 6.

Reviewer #3: Minor comments/questions:

1. By RNA-seq analysis the proportion of virus-host chimeric reads were much higher in n=9 MmuPV1-induced tumor tissues (1.9-7) compared to n=3 MmuPV-1-infected tumor-free skin (0.6-1,3). Are the results statistically significant?

2. The numbers of integrated mucosal high-risk HPV copies correlate with severity of cervical disease and progression. MmuPV1 infection led to development of SCCs in ear skin (ref 39). Did the authors have a chance to look into integration sites in MmuPV1-induced SCCs or their premalignant precursors on ear skin? Do they think that higher levels of integrated virus may correlate with increasing malignancy in virus-induced skin cancer, similar to the cervix?

3. When did the authors harvest the papillomas, the latest 6 months post-infection? As immunodeficient FoxN1 nude mice were used, it is unlikely that the papillomas will regress. Do the authors expect more (new) genes affected by integration in the aging mouse, e.g genes responsible for cancer growth and development, after 1 or 2 years.

4. Could the authors please state the number of animals analyzed in the Material and Methods section.

PLOS authors have the option to publish the peer review history of their article (what does this mean?). If published, this will include your full peer review and any attached files.

Reviewer #1: No

Reviewer #2: No

Reviewer #3: No

Figure Files:

Data Requirements:

Reproducibility:

References:

---

## [Editor Report · Decision Letter 1]

19 Jul 2021

Dear Dr. Zheng,

We are pleased to inform you that your manuscript 'Mouse papillomavirus type 1 (MmuPV1) DNA is frequently integrated in benign tumors by microhomology-mediated end-joining' has been provisionally accepted for publication in PLOS Pathogens.

Best regards,

Richard BS Roden

Associate Editor

PLOS Pathogens

Alison McBride

Section Editor

PLOS Pathogens

Kasturi Haldar

Editor-in-Chief

PLOS Pathogens

orcid.org/0000-0001-5065-158X

Michael Malim

Editor-in-Chief

PLOS Pathogens

orcid.org/0000-0002-7699-2064
---

## [Editor Report · Acceptance letter]

28 Jul 2021

Dear Dr. Zheng,

We are delighted to inform you that your manuscript, "Mouse papillomavirus type 1 (MmuPV1) DNA is frequently integrated in benign tumors by microhomology-mediated end-joining," has been formally accepted for publication in PLOS Pathogens.

Best regards,

Kasturi Haldar

Editor-in-Chief

PLOS Pathogens

orcid.org/0000-0001-5065-158X

Michael Malim

Editor-in-Chief

PLOS Pathogens

orcid.org/0000-0002-7699-2064